# Structure and tethering mechanism of dynein-2 intermediate chains in intraflagellar transport

Aakash G Mukhopadhyay [ID] [1,2], Katerina Toropova [ID] [1,2], Lydia Daly [ID] [2,5], Jennifer N Wells [ID] [2,6], Laura Vuolo[3], Miroslav Mladenov [ID] [2,7], Marian Seda [ID] [4], Dagan Jenkins [ID] [4], David J Stephens [ID] [3] & Anthony J Roberts [ID] [1,2✉]

## Abstract

**Dynein-2 is a large multiprotein complex that powers retrograde intraflagellar transport (IFT) of cargoes within cilia/flagella, but the molecular mechanism underlying this function is still emerging. Distinctively, dynein-2 contains two identical force-generating heavy chains that interact with two different intermediate chains (WDR34 and WDR60). Here, we dissect regulation of dynein-2 function by WDR34 and WDR60 using an integrative approach including cryo-electron microscopy and CRISPR/Cas9-enabled cell biology. A 3.9 Å resolution structure shows how WDR34 and WDR60 use surprisingly different interactions to engage equivalent sites of the two heavy chains. We show that cilia can assemble in the absence of either WDR34 or WDR60 individually, but not both subunits. Dynein-2-dependent distribution of cargoes depends more strongly on WDR60, because the unique N-terminal extension of WDR60 facilitates dynein-2 targeting to cilia. Strikingly, this N-terminal extension can be transplanted onto WDR34 and retain function, suggesting it acts as a flexible tether to the IFT "trains" that assemble at the ciliary base. We discuss how use of unstructured tethers represents an emerging theme in IFT train interactions.**

**Keywords** Cilia; Dynein; Intraflagellar Transport; Microtubule
**Subject Categories** Cell Adhesion, Polarity & Cytoskeleton; Membranes & Trafficking; Structural Biology

## Introduction

Dynein motor complexes power diverse microtubule-based motilities. Intermediate chains (ICs) are a ubiquitous feature of multimeric dyneins, including cytoplasmic dynein-1, which transports cargoes in the cell interior (Reck-Peterson et al, 2018; Roberts et al, 2013); outer-arm dynein and dynein-f, which are involved in

axonemal beating (King, 2016); and dynein-2, which is vital for the assembly and functions of cilia/flagella (Vuolo et al, 2020). Consequently, mutations in dynein ICs are associated with a wide range of human disorders spanning microcephaly (Ansar et al, 2019), primary ciliary dyskinesia (Guichard et al, 2001), and short-rib thoracic dysplasia (Schmidts et al, 2013; McInerney-Leo et al, 2013). Understanding IC mechanism is thus critical for deciphering the molecular basis of dynein function and associated disease states.

ICs bind directly to the force-generating heavy chains (HCs) of dyneins, specifically to the elongated "tail" which protrudes from the AAA+ motor domain and mediates oligomerization and cargo binding (Tynan et al, 2000; Zhang et al, 2017) (Fig. 1A). Each IC consists of a β-propeller domain, which binds to the HC, and an N-proximal region, which binds to light chains (Ma et al, 1999; Williams et al, 2005; Lo et al, 2001; Susalka et al, 2002). In dynein-1, an N-terminal α-helix of the IC mediates binding to the regulatory proteins p150 and Nudel (Vaughan and Vallee, 1995; Karki and Holzbaur, 1995; McKenney et al, 2011). Typically, the oligomeric state of the IC matches the oligomeric state of the associated HC e.g., cytoplasmic dynein-1 contains a homodimeric IC and HC, whereas outer-arm dynein and dynein-f contain heterodimeric ICs and HCs (Reck-Peterson et al, 2018; King, 2016; Roberts, 2018).

An exception to this pattern is found in dynein-2 (also known as intraflagellar transport dynein or dynein-1b), which contains two identical HCs that partner with two different ICs, WDR34 and WDR60 (Rompolas et al, 2007; Patel-King et al, 2013; Asante et al, 2014; Hamada et al, 2018; Toropova et al, 2019). Dynein-2 is the essential motor for retrograde intraflagellar transport (IFT) (Pazour et al, 1999; Porter et al, 1999; Signor et al, 1999), the conserved process in which polymeric IFT "trains" translocate cargoes along the ciliary microtubules (Vuolo et al, 2020). Each IFT train is a co-polymer of IFT-A and -B complexes and kinesin-2 and dynein-2 motors (Nakayama and Katoh, 2020; Taschner and Lorentzen, 2016). Dynein-2 assembles with IFT trains in an inhibited conformation at the transition zone (TZ, the diffusion barrier at the ciliary base); is carried under the power of kinesin-2 to the ciliary tip (anterograde IFT); and is then activated to power transport back toward the cell body via the TZ (retrograde IFT) (Webb et al, 2020; Jordan and Pigino, 2021). Structural information is available for the dynein-2 complex in its inhibited state at 4.5–38 Å resolution, revealing a highly asymmetric structure that appears

[1]Sir William Dunn School of Pathology, University of Oxford, Oxford, UK. [2]Institute of Structural and Molecular Biology, Department of Biological Sciences, Birkbeck, University of London, London, UK. [3]Cell Biology Laboratories, School of Biochemistry, University of Bristol, Bristol, UK. [4]UCL Great Ormond Street Institute of Child Health, University College London, London, UK. [5]Present address: Randall Centre of Cell & Molecular Biophysics, King's College London, London, UK. [6]Present address: MRC London Institute of Medical Sciences (LMS), London, UK. [7]Present address: Cellular Signalling and Cytoskeletal Function Laboratory, The Francis Crick Institute, London, UK.
✉E-mail: anthony.roberts@path.ox.ac.uk

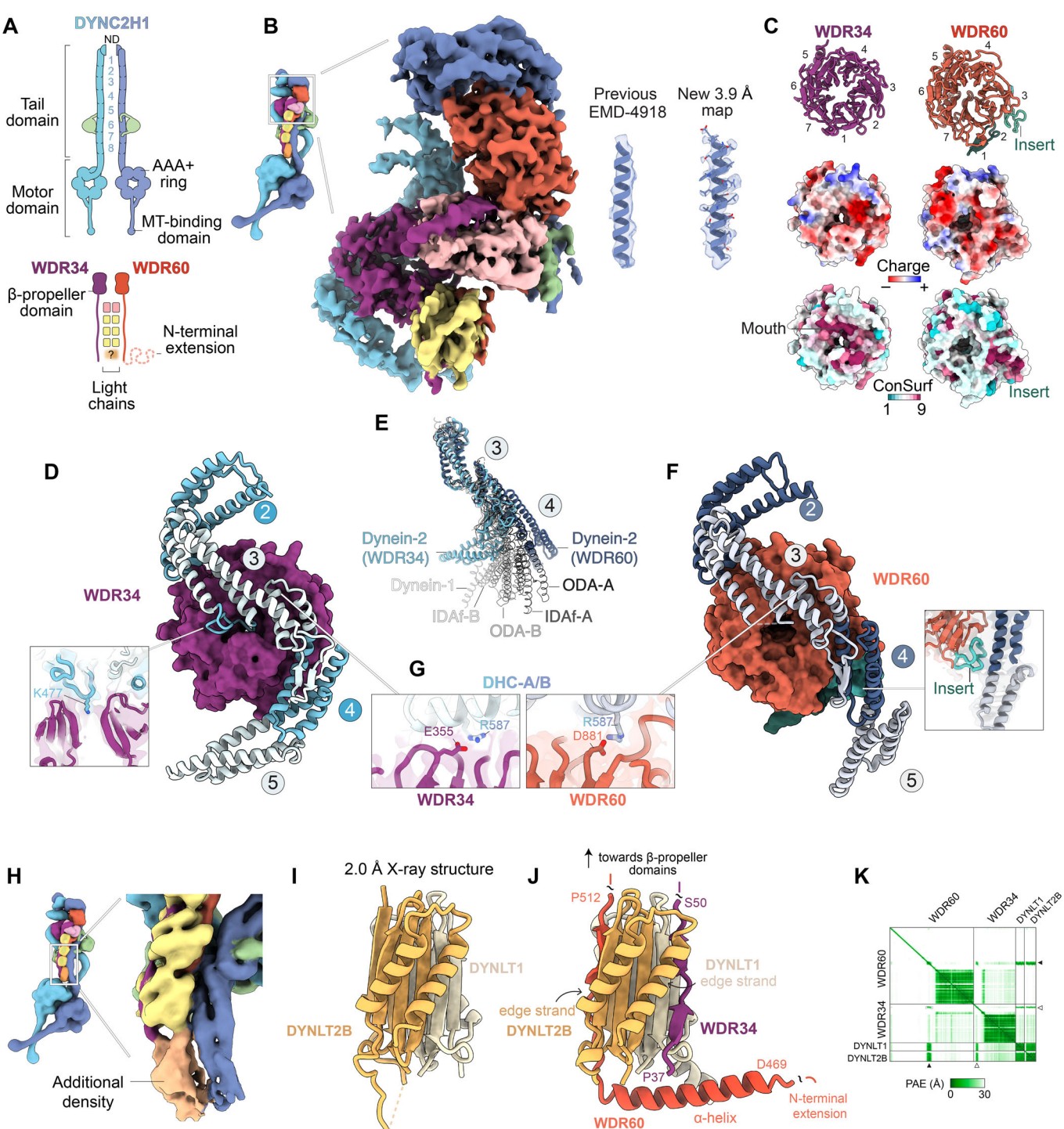

tailored to the structure of the anterograde IFT-B polymer (Jordan et al, 2018; Toropova et al, 2019; van den Hoek et al, 2022; Lacey et al, 2023). Congruently, biochemical assays have suggested multiple points of contact between dynein-2 and IFT-B (Vuolo et al, 2018; Hamada et al, 2018; Tsurumi et al, 2019; Zhu et al, 2021; Hiyamizu et al, 2023a, 2023b). At the present structural resolutions, it has been unclear how the two ICs are integrated into the dynein-2 complex and how they contribute to its architecture and function at the amino acid level, because current models have been limited to backbone atoms only.

Cell biology studies in which WDR34 or WDR60 have been individually mutated, knocked down, or knocked out have shed light on the importance of these subunits for ciliogenesis, dynein-2 assembly, cargo localization, and transition zone integrity, while revealing that the N-terminal region of WDR34 can exert a dominant negative effect (Asante et al, 2013; Huber et al, 2013; McInerney-Leo et al, 2013; Schmidts et al, 2013; Asante et al, 2014; Blisnick et al, 2014; Wu et al, 2017; Hamada et al, 2018; Jensen et al, 2018; Vuolo et al, 2018; Tsurumi et al, 2019; Shak et al, 2023). It has

**Figure 1. WDR34 and WDR60 interact differently with DYNC2H1.**

(A) Diagram of subunits within the dynein-2 complex. The two identical copies of DYNC2H1 are colored different shades of blue for distinction, α-helical bundles within the tail domain are numbered (1–8). DYNC2LI1; green. WDR34; purple. WDR60; red. DYNLRB1; pink. DYNLL1; yellow. Orange blur and question mark reflect undefined stoichiometry of DYNLT1 and DYNLT2B. (B) Composite cryo-EM map of WDR34 and WDR60 within the dynein-2 complex, following focussed 3D refinement (Appendix Fig. S1). Enlargement shows improvement in map quality compared to EMD-4918. (C) Top row; Ribbon diagram of WDR34 and WDR60 β-propeller domains, with blades numbered (1–7). Insert within blade 3 of WDR60 is highlighted. Middle row; surface representation of WDR34 and WDR60 colored by Coulombic electrostatic potential. Bottom row; surface representation of WDR34 and WDR60 colored by ConSurf score (Yariv et al, 2023), variable residues (score 1) in cyan, conserved residues (score 9) in magenta. (D) Interface between WDR34 β-propeller domain (purple surface representation) and DYNC2H1 (ribbon representation, α-helical bundles numbered). Inset; a loop from bundle 3 of DYNC2H1 positions a lysine residue (K477) in the pore of the WDR34 β-propeller. (E) Heavy chain conformation in dynein-2, outer-arm dynein (PDB 8GLV), dynein-f (PDB 8GLV) (Walton et al, 2023), and dynein-1 (PDB 7Z8F) (Chaaban and Carter, 2022), following alignment on α-helical bundle 3. (F) Interface between WDR60 β-propeller domain (red surface representation) and DYNC2H1 (ribbon representation, α-helical bundles numbered). Inset; WDR60 insert stabilizes DYNC2H1 in a "straight" conformation. (G) WDR34 and WDR60 use an equivalent acidic residue (E355 and D881 respectively) to interact with R587 of DYNC2H1. (H) Additional ambiguous density at the base of the dynein-2 light chain tower (Toropova et al, 2019). (I) X-ray crystallography structure of DYNLT1-DYNLT2B heterodimer. (J) AF Multimer model of DYNLT1-DYNLT2B interaction with WDR34 (residues P37–50) and WDR60 (residues D469–P512). (K) Predicted aligned error (PAE) plot associated with (J). Note the high-confidence prediction (low PAE) for the interface between DYNLT1-DYNLT2B and WDR34 (white arrowhead)) and WDR60 (black arrowhead). Source data are available online for this figure.

been challenging to visualize the localization of the dynein-2 heavy chain in living mammalian cells, hampering insights into which stage/s of the dynein-2 transport cycle are affected by perturbations in WDR34 or WDR60, and why. An elegant study in *C. elegans* identified a WDR60 homolog (C27F2.1) as important for dynein-2 attachment to IFT trains and passage through the TZ (De-Castro et al, 2022). Intriguingly, it is unclear if a second dynein-2 IC exists in this system, or if the *C. elegans* WDR60 homolog functions as a homodimer (Higashida and Niwa, 2022). Here, to address the relative contributions of WDR34 and WDR60 to dynein-2 structure, dynamics, and mechanism, we took an integrative approach combining cryo-EM, crystallography, CRISPR/Cas9, and live-cell imaging of dynein-2 in mammalian cells.

## Results

Elucidating how WDR34 and WDR60 are integrated into the dynein-2 complex at amino acid resolution required several methodological advances. First, we improved the biochemical homogeneity of the dynein-2 preparation by removing redundant subunit isoforms from the expression construct (Toropova et al, 2019; Hamada et al, 2018). This yielded a streamlined construct expressing eight subunits, each under the control of the *polH* promoter for insect cell expression (Appendix Fig. S1). The subunits in the final construct are the dynein-2 heavy chain (DYNC2H1), two intermediate chains (WDR60/DYNC2I1 and WDR34/DYNC2I2), light intermediate chain (DYNC2LI1), and single isoforms of the four dynein-2 light chains (DYNLRB1, DYNLL1, DYNLT1, and DYNLT2B/TCTEX1D2). Second, we collected a 12,975 micrograph cryo-EM dataset of the purified dynein-2 complex using a K3 direct detector camera with aberration-free image shift (AFIS) acquisition. Third, we used multiple rounds of mask design to identify optimal regions for focussed 3D refinement of WDR34 and WDR60 and their interface with DYNC2H1 (Appendix Fig. S1F). Whereas in the previous cryo-EM map (EMD-4918) α-helices of DYNC2H1 interacting with WDR34 and WDR60 appear as tubular densities, permitting modeling of the backbone atoms only, in the new map many side chains are resolved (Fig. 1B), enabling an atomic model.

### WDR34 and WDR60 interact differently with DYNC2H1

Our cryo-EM structure reveals that WDR34 and WDR60 use surprisingly different interactions to engage equivalents sites on

DYNC2H1 (Fig. 1B–F). WDR34 and WDR60 both contain a C-terminal β-propeller domain, with 22% sequence identity. Each β-propeller has seven radially arranged blades, the seventh of which is distorted in WDR60 giving the propeller a more elliptical appearance (Fig. 1C, top row). The DYNC2H1 tail comprises an N-terminal homodimerization domain (ND) followed by eight α-helical bundles (Fig. 1A). The WDR34 and WDR60 β-propeller domains each engage a copy of DYNC2H1, principally at bundles 3 and 4. Notably, a loop from bundle 3 of DYNC2H1 positions a lysine residue (K477) directly into the pore of the WDR34 β-propeller domain (Fig. 1D), whose mouth is lined by highly conserved residues (Fig. 1C, bottom row). Conversely, the equivalent interaction is lacking in WDR60: the loop from bundle 3 of DYNC2H1 is disordered and the residues of WDR60 around the mouth of the pore are poorly conserved. Instead, WDR60 uses a conserved insert within blade 3 of its β-propeller (Fig. 1C) to contact bundle 4 of DYNC2H1 (Fig. 1F). This WDR60 insert, which is absent in WDR34, stabilizes the bundle 3/4 junction of DYNC2H1 in a "straight" conformation. In contrast, the copy of DYNC2H1 associated with WDR34 is bent (Fig. 1D *cf*. F). These straight and bent conformers of DYNC2H1 represent extreme ends of the range of heavy chain conformations observed in other dyneins (Fig. 1E). The only notable shared interaction that WDR34 and WDR60 use to engage DYNC2H1 involves E355 (WDR34)/ D881 (WDR60) contacting R587 of DYNC2H1 (Fig. 1G). DYNC2H1 mutation R587C is associated with severe short-rib thoracic dysplasia (Merrill et al, 2009). In summary, WDR34 and WDR60 use strikingly different interactions to engage the two copies of DYNC2H1 and stabilize them in different conformations.

### DYNLT1 and DYNLT2B form a heterodimer between WDR34 and WDR60

The regions of WDR34 and WDR60 N-proximal to the β-propeller domains are held together by a tower of dimeric light chains (one DYNLRB1 homodimer and three DYNLL1 homodimers) (Toropova et al, 2019). At the base, there is an additional density in the cryo-EM map that likely corresponds to DYNLT1/DYNLT2B (Fig. 1H), but which is too weak to permit model building. Previous data show that DYNLT1 and DYNLT2B interact, making a heterodimer the most plausible stoichiometry (Hamada et al, 2018), but a dimer of dimers has not been ruled out (Braschi et al, 2022). We therefore purified the DYNLT1-DYNLT2B complex,

used SEC-MALS to show it is a heterodimer (mass of 28.1 kDa) (Appendix Fig. S2A–C), and determined its structure at 2 Å resolution using X-ray crystallography (Appendix Fig. S2D). The structure reveals a domain-swapped heterodimer (Fig. 1I) that accounts for the unfilled density in the cryo-EM map (Appendix Fig. S2E). We then used AlphaFold (AF) Multimer to generate a high-confidence prediction of the regions of WDR34 and WDR60 that interact with the DYNLT1-DYNLT2B heterodimer (Fig. 1J,K). In this model, WDR34 residues P37-S50 bind an edge strand of DYNLT1, whereas WDR60 residues D469–P512 bind an edge strand of DYNLT2B, bend through 90 degrees, and form an α-helix that wraps around DYNLT1-DYNLT2B (Fig. 1J). The DYNLT1-DYNLT2B-binding region of WDR60 is in precise agreement with that mapped biochemically (Hamada et al, 2018), providing experimental validation for the model. N-terminal to the DYNLT1-DYNLT2B binding site, WDR60 contains a further 470 amino acids that are not resolved in the cryo-EM map, suggestive of high flexibility, but whose functions we investigate below using cell biology. In summary, these data suggest that the N-proximal regions of WDR34 and WDR60 are held together by a DYNLT1-DYNLT2B heterodimer, contrasting with the other dynein-2 light chains which are homodimers.

## Cilia can assemble in the absence of either WDR34 or WDR60, but not both

To dissect the relative contributions of WDR34 and WDR60 to cilia formation and IFT dynamics, we used CRISPR-Cas9 to knock out WDR34, WDR60, or both subunits in IMCD-3 cells; a mouse kidney cell line which forms cilia upon serum starvation (Fig. 2A–C; Appendix Fig. S3) (Mukhopadhyay et al, 2010; Ye et al, 2018). We used immunofluorescence (IF) microscopy to assess the impact on ciliary length, using acetylated tubulin as a marker for the ciliary axoneme and gamma-tubulin as a marker of the basal body (Fig. 2C). This analysis revealed that axoneme length was not affected by loss of either WDR34 or WDR60 individually (Fig. 2B,C). However, the double KO lacking both WDR34 and WDR60 displayed short, stumpy cilia with an average length of $1.5 \pm 0.1$ μm (mean ± SEM), suggestive of severely impaired dynein-2 function. Indeed, the impact of knocking out both WDR34 and WDR60 was akin to that of knocking out the dynein-2 motor subunit, DYNC2H1, which we performed in IMCD-3 cells for comparison (Fig. 2B,C; Appendix Fig. S3).

We next assessed ciliary cargo distribution—a more sensitive readout of dynein-2 function—in the WDR34 KO, WDR60 KO, and double KO backgrounds. Imaging of IFT88 (a core subunit of IFT-B) tagged with mNeonGreen (NG3IFT88) revealed a progressively severe accumulation of IFT88 within cilia when WDR34, WDR60, or both subunits were lacking (Fig. 2D,E). The accumulation of IFT-B within cilia is a hallmark of impaired retrograde IFT, in which the flux of IFT trains to the cilia tip exceeds that of dynein-2-driven retrograde trains back to the cell body (Vuolo et al, 2020; Hamada et al, 2018; Tsurumi et al, 2019; Hiyamizu et al, 2023b).

Smoothened (Smo), a transmembrane receptor involved in Hedgehog signaling (Chen and Jiang, 2013), also displayed a ciliary localization pattern suggestive of progressively severe defects in retrograde IFT in WDR34, WDR60, and double KO cells. Smo is canonically shuttled out of cilia in basal conditions by retrograde

IFT and accumulates in cilia only when the Hedgehog pathway is activated (or when cells are treated with Smoothened agonist; SAG) (Fig. 2F,G) (Keady et al, 2012; Eguether et al, 2018). In WDR34 KO cells, Smo was present within cilia even in basal conditions, albeit very rarely, while WDR60 KO cells frequently displayed Smo-positive cilia. Double KO cells in basal conditions exhibited Smo-positive cilia almost as frequently as control cells treated with SAG (Fig. 2F,G).

Together, these single and double KO cells indicate that (i) there is sufficient functionality in the dynein-2 pathway to assemble normal length cilia in the absence of WDR34 or WDR60 individually, but not when both subunits are removed, suggestive of a threshold amount of retrograde IFT being required for proper ciliogenesis (Engel et al, 2012); (ii) at the more sensitive level of ciliary cargo distribution, individual loss of WDR34 or WDR60 is sufficient to cause defects, with loss of WDR60 having a more severe effect.

## Impact of WDR34 or WDR60 deletion on IFT dynamics

We next asked which aspect/s of IFT train dynamics are modulated by the deletion of WDR34 or WDR60, using NG3IFT88 to track IFT train movement by live-cell TIRF microscopy (Fig. 3A) (Liew et al, 2014; Ye et al, 2018). In control cells, IFT trains could be readily seen moving to the tip of the cilium and back to the base (Fig. 3B, top row). In double KO cells lacking WDR34 and WDR60, IFT was scarcely detectable (Fig. 3B, bottom row). In cells lacking WDR34 or WDR60 individually, IFT was clearly visible, but the dynamics appeared to be affected (Fig. 3B, middle rows).

We used kymograph analysis to quantify the movements and categorize them into the kinesin-2 direction (anterograde IFT) or dynein-2 direction (retrograde IFT) (Fig. 3C) (Mangeol et al, 2016). Loss of WDR34 or WDR60 reduced the frequency of movements in the dynein-2 direction compared to control cells, with loss of WDR60 having a more severe effect (Fig. 3C). Similarly, loss of WDR34 or WDR60 reduced the velocity of movements in the dynein-2 direction, with loss of WDR60 again having a more severe impact (Fig. 3C). There was no effect on the velocity in the kinesin-2 direction, while the frequency of movements in the kinesin-2 direction was subtly reduced (Fig. 3C), likely due to perturbed recycling of IFT train components back to the ciliary base (Wingfield et al, 2017). In summary, these live-cell imaging data reveal that loss of WDR34 or WDR60 reduces (but does not abolish) both the frequency and velocity of dynein-driven IFT, markedly so in the case of WDR60.

## Loss of WDR60 and WDR34 impairs targeting of dynein-2 into cilia

We next investigated the mechanism underlying the perturbed IFT dynamics in the absence of WDR34 and WDR60. Western blotting showed that the stability of DYNC2H1 is not affected by the loss of WDR34 or WDR60 (Appendix Fig. S4A), suggesting that the defects arise from defects in dynein-2 regulation rather than reduced levels of the motor subunit. Two plausible explanations for perturbed retrograde IFT are (1) a failure of the dynein-2 motor subunit, DYNC2H1, to load properly onto IFT trains at the ciliary base, resulting in reduced levels of DYNC2H1 per cilium

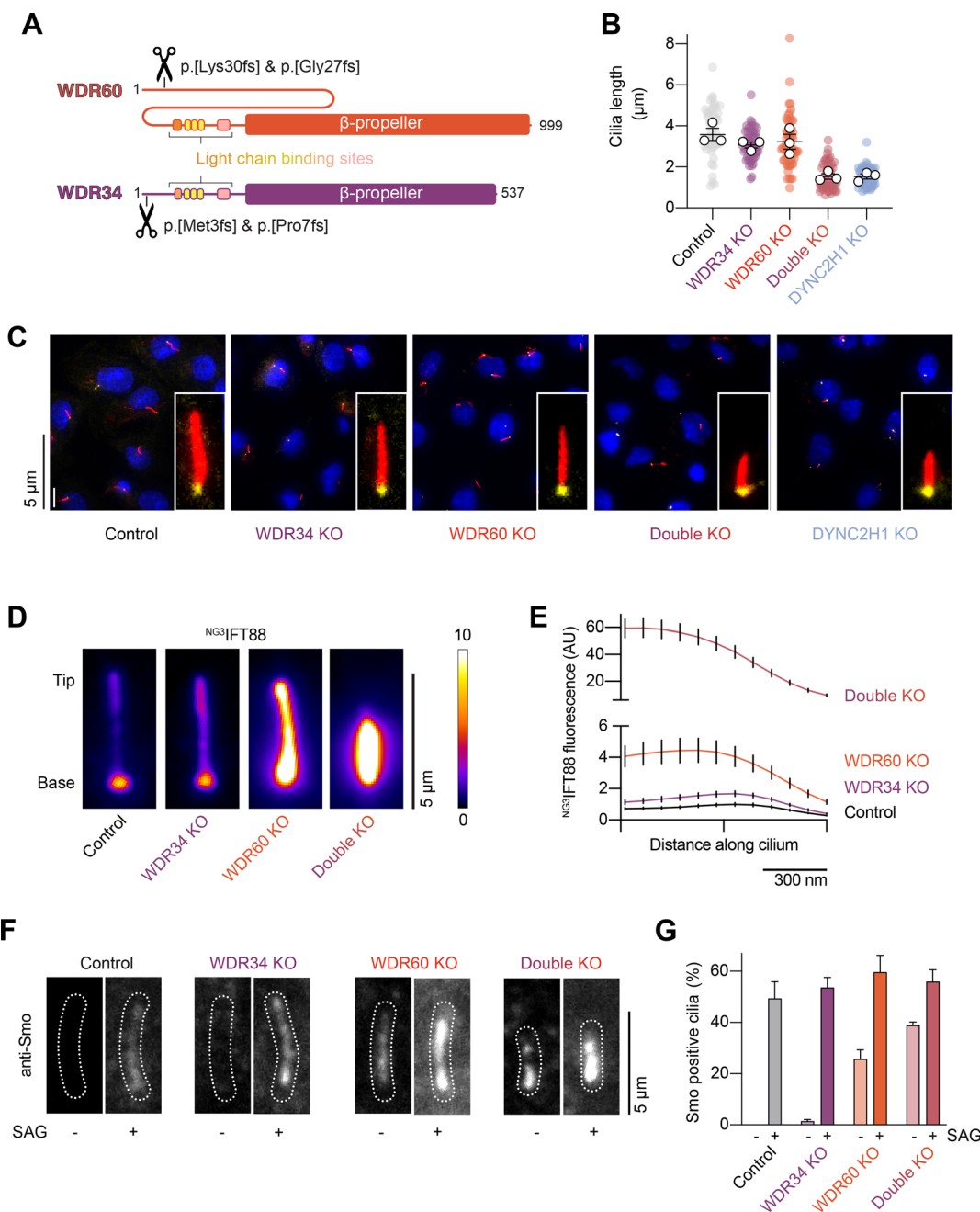

**Figure 2. Cilia can assemble in the absence of either WDR34 or WDR60, but not both.**

(A) Sequence diagram of mouse WDR60 and WDR34. Scissors represent the location of CRISPR-Cas9 mediated genome editing. See also Appendix Fig. S3.
(B) Quantification of cilia length in the indicated IMCD-3 cell lines from three technical replicates. Colored circles; individual data points. White circles; average from each separate experiment. Lines; mean (±SEM). 45 control, 50 WDR34 KO, 60 WDR60 KO, 49 double KO, and 50 DYNC2H1 KO cilia measured. WDR34 KO and WDR60 KO cilia length is not significantly altered vs control (one-way ANOVA followed by Kruskal–Wallis test, $p > 0.05$). Double KO cilia length is reduced vs control ($p < 0.0001$). Cilia length for double KO vs DYNC2H1 KO is not significantly different ($p > 0.05$). (C) Representative images and enlargements of IMCD-3 cilia immunofluorescently labeled for acetylated tubulin (red) and gamma-tubulin (yellow) in the indicated cell lines. DAPI (blue) is used to stain nuclei. Scale bar 5 μm. (D) Time-averaged images of [NG3]IFT88 fluorescence in the indicated cell lines, colored according to the fire scale. Scale bar 5 μm. (E) Plots of average [NG3]IFT88 fluorescence intensity from line scans along the cilium length, aligned at the ciliary tip. Values are normalized relative to the control cilia peak value. Traces show mean intensity ± SEM; 46 control, 51 WDR34 KO, 54 WDR60 KO, 28 double KO cilia analyzed from three technical replicates. Scale bar 300 nm. (F) Representative images of indicated cell lines in the presence or absence of SAG, fixed and stained for Smo. Cilia (dashed outlines) identified by [NG3]IFT88 signal. Scale bar 5 μm. (G) Percentage Smo-positive cilia (mean ± SEM) of indicated cell lines in the presence or absence of SAG, from three technical replicates. SAG untreated (154 control, 138 WDR34 KO, 163 WDR60 KO, 248 double KO) and SAG treated (129 control, 168 WDR34 KO, 170 WDR60 KO, 298 double KO) cilia analyzed. In the absence of SAG, the percentage of Smo-positive cilia is significantly increased for WDR60 KO and double KO vs control (one-way ANOVA followed by Kruskal–Wallis test, $p < 0.0001$). In the presence of SAG, the percentage of positive cilia is not significantly altered in different knockout cell lines vs control. ($p > 0.05$). Percentage Smo-positive cilia for control (SAG treated) vs double KO (SAG untreated) is not significantly different ($p > 0.05$). Source data are available online for this figure.

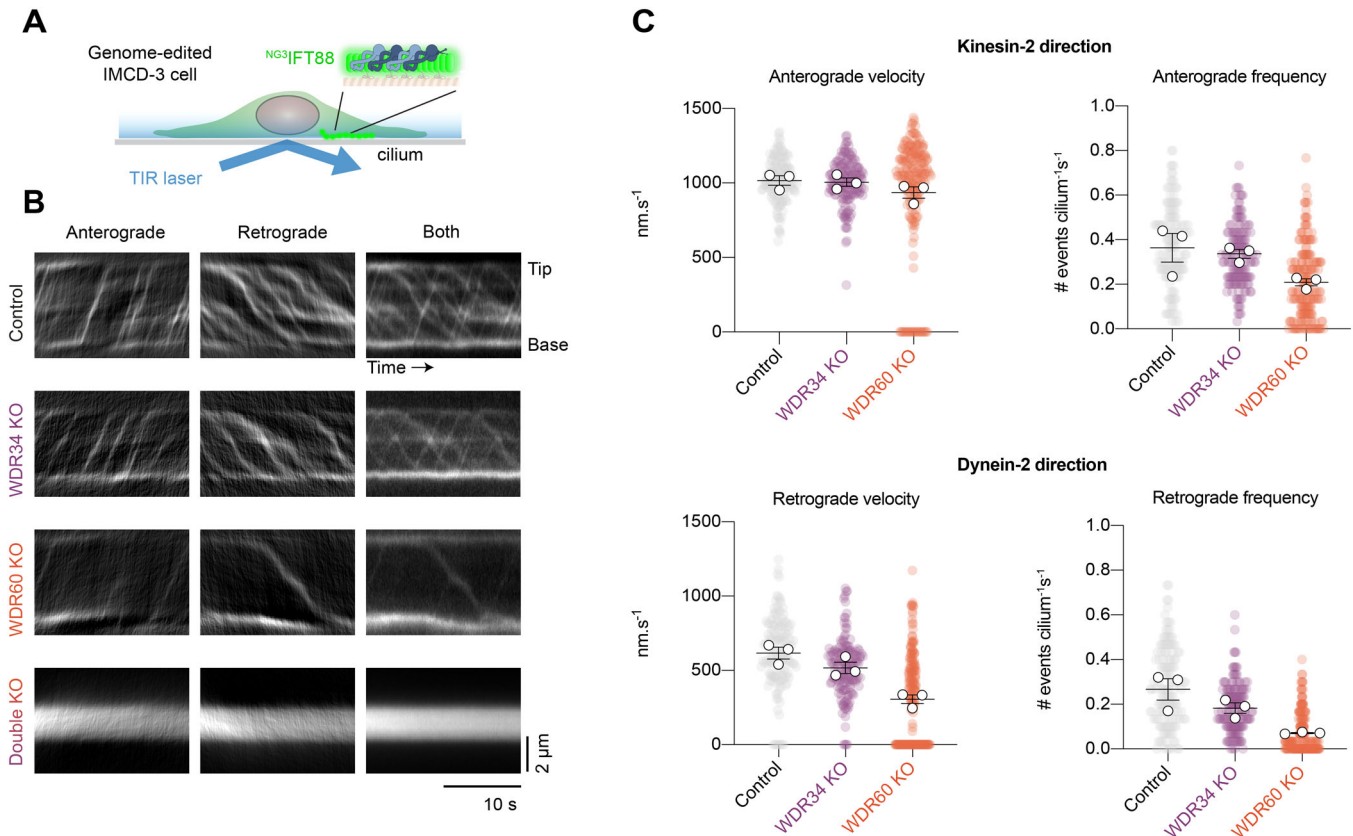

**Figure 3.  Impact of WDR34 and WDR60 deletion on IFT dynamics.**

(**A**) Schematic showing experimental setup to image IFT dynamics in [NG3]IFT88 IMCD-3 cilia by TIRF microscopy. (**B**) Representative kymographs of anterograde and retrograde [NG3]IFT88 movement in control, WDR34 KO, WDR60 KO, and double KO cilia. Scale bars: Vertical 2 μm; Horizontal 10 s. (**C**) Quantification of [NG3]IFT88 velocity and frequency in anterograde (kinesin) and retrograde (dynein) directions in the indicated cell lines from three technical replicates. Colored circles; individual data points. White circles; average from each separate experiment. Lines; mean (±SEM). 140 control, 154 WDR34 KO, and 190 WDR60 KO cilia were analyzed. Anterograde velocities were unaffected in WDR34 KO and WDR60 KO cilia compared to control ($p > 0.05$, one-way ANOVA followed by Kruskal–Wallis test). The anterograde frequency of WDR60 KO cilia was reduced, ($p < 0.0001$), while WDR34 KO anterograde frequency was unaffected ($p > 0.05$). Retrograde velocities were significantly reduced in WDR34 KO cilia ($p < 0.001$) and WDR60 KO cilia ($p < 0.0001$). Retrograde frequency was reduced in both WDR34 KO ($p < 0.005$) and WDR60 KO cilia ($p < 0.0001$). Source data are available online for this figure.

(De-Castro et al, 2022) or (2) a failure of the dynein-2 motor subunit to activate at the ciliary tip, resulting in ciliary accumulation of DYNC2H1. To test these possibilities, we stably expressed DYNC2H1 with an mScarlet tag ([mScar]DYNC2H1) for visualization (Appendix Fig. S4B). We made use of the PiggyBac transposon system for these experiments, as the enormous size of the DYNC2H1 open reading frame (13 kb) precludes the use of lentivirus. We found that the amount of DYNC2H1 signal in the cilium progressively decreased in WDR34, WDR60, and double KO cells (Fig. 4A,B). This reduction in DYNC2H1 per cilium was reciprocally matched by the accumulation of [NG3]IFT88, which we visualized in the same cell line for comparison (Fig. 4A,B). Whereas [NG3]IFT88 displayed accumulation at the ciliary tip in WDR60 KO cells, no such accumulation was observed within the weaker DYNC2H1 signal. We confirmed that the reduced level of DYNC2H1 in WDR34, WDR60 and double KO cells is also observed for the endogenous protein using immunofluorescence (Appendix Fig. S4C). Taken together, these data suggest that the motility defects in cells lacking WDR34 and WDR60 arise from impaired targeting of dynein-2 into cilia.

## The N-terminal extension of WDR60 is a modular element that tethers dynein-2 to IFT-B

Why is the dynein-2 function more dependent on WDR60 than WDR34? A strong candidate is the presence of the long N-terminal extension in WDR60, which previous studies have found to bind to the IFT-B subunit IFT54 (Hiyamizu et al, 2023b) and localize to cilia independently of the dynein-2 heavy chain (De-Castro et al, 2022). We analyzed the WDR60 N-terminal extension using ConSurf (Yariv et al, 2023), which revealed four conserved patches of amino acids N-terminal to the DYNLT1-DYNLT2B binding site (Fig. 5A). We then used AF Multimer to screen for potential binding partners within the IFT-A and -B subunits of the IFT train (using the full-length WDR60 sequence to avoid any bias toward the N-terminal extension). A high-confidence interaction was predicted between residues 375–388 of the WDR60 N-terminal extension and the CH domain of IFT54 (Fig. 5B), in excellent agreement with prior visual immunoprecipitation (VIP) assays (Hiyamizu et al, 2023b). We confirmed this direct interaction using a pull-down assay with purified proteins (Appendix Fig. S5A).

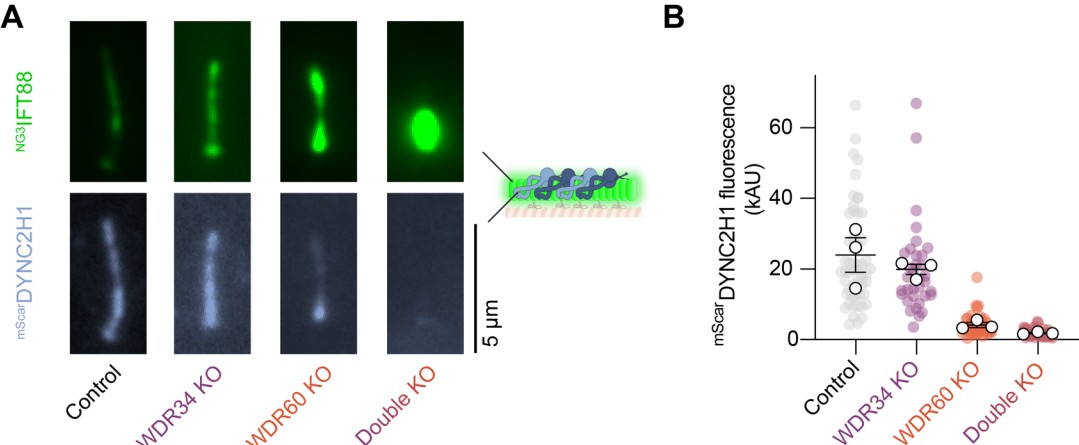

**Figure 4. Importance WDR34 and WDR60 in dynein-2 targeting into cilia.**

(A) Representative time-averaged images of ^NG3^IFT88 and ^mScar^DYNC2H1 signal in the same cilium of the indicated cell lines. Scale bar 5 µm. (B) Quantification of ^mScar^DYNC2H1 signal in the cilia of the indicated cell lines from three technical replicates. Colored circles; individual data points. White circles; average from each separate experiment. Lines; mean (±SEM). 55 control, 34 WDR34 KO, 46 WDR60 KO, and 44 double KO cilia were analyzed. ^mScar^DYNC2H1 intensity within cilia was significantly reduced in WDR60 KO and double KO compared to control ($p < 0.0001$). ^mScar^DYNC2H1 signal in double KO cells was further reduced in comparison to WDR60 KO ($p < 0.05$). Source data are available online for this figure.

A second high-confidence interaction was detected between residues 407–433 of the WDR60 N-terminal extension and the C-terminal TPR domain of IFT80 (Fig. 5C). This previously undescribed interaction is consistent with IFT80 co-immunoprecipitating with WDR60, as detected by mass spectrometry (Hiyamizu et al, 2023b). As a direct binding assay was not possible with IFT80, owing to the poor solubility of purified human IFT80, this interaction remains putative. To examine if and how the WDR60 N-terminal extension relates to dynein-2's attachment to anterograde IFT trains, we built a structural model combining our dynein-2 cryo-EM structures, DYNLT1-DYNLT2B crystal structure, and AF models, together with the cryo-ET model of the anterograde IFT train (Lacey et al, 2023) (Fig. 5D). This revealed that, following the WDR60 α-helix that wraps around DYNLT1-DYNLT2B (Fig. 1J), the N-terminal extension of WDR60 is optimally positioned for residues 407–433 to contact the exposed IFT80 C-terminal TPR domain and places residues 375–388 within contact distance of the IFT54 CH domain, which lies at the end of a long flexible linker. Another reported interaction between dynein-2 and IFT-B involving DYNC2LI1 and the IFT54 coiled-coil domain is not physically compatible with the anterograde IFT train structure, suggesting it could be used elsewhere in the IFT cycle (Hiyamizu et al, 2023b). Together, these data suggest that the N-terminal extension of WDR60 forms a flexible tether between dynein-2 and the anterograde IFT train.

To test the importance of the WDR60 N-terminal extension in dynein-2 function, we generated constructs lacking the N-terminal extension (ΔN470) or the N-terminal extension and light chain binding sites (ΔN630) of WDR60. We exploited our double KO cells as a sensitive background in which to assess the functionality of WDR60. Whereas expression of WT WDR60 rescued the severe phenotypes of double KO cells (stumpy cilia and strong IFT88 accumulation), the construct lacking the N-terminal extension (ΔN470) was completely unable to rescue, akin to the ΔN630 construct (Fig. 5E–G). We observed the same trend when the

WDR60 constructs were expressed in single WDR60 KO cells (Appendix Fig. S5B–D). These data suggest that the N-terminal extension of WDR60 is crucial for proper dynein-2 function.

If the WDR60 N-terminal extension serves as a flexible tether to the IFT train, it may be possible to transplant it onto WDR34 and retain its function. To test this idea, we generated a chimeric construct comprising WDR60 residues 1–470 fused with the full-length WDR34 sequence (Fig. 5H). Strikingly, whereas expression of WDR34 rescued the short cilia phenotype but not the IFT88 accumulation of double KO cells, consistent with our single KO data, expression of the chimeric construct rescued both cilia length and IFT88 accumulation to a level comparable to WDR60 itself (Fig. 5H,I). Thus, the N-terminal extension of WDR60 is a modular element that can be transplanted onto WDR34 to rescue function.

## Discussion

Here we have used structural and cell biology approaches to shed new light on how dynein-2, the ubiquitous motor for retrograde IFT, is regulated by its distinctive heterodimeric ICs WDR34 and WDR60. Our cryo-EM structures show that the WDR60 β-propeller is distorted relative to WDR34 and uses a unique insert in its third blade to stabilize DYNC2H1 in a straight confirmation. In contrast, WDR34 uses a classic β-propeller mode interaction to engage DYNC2H1, accommodating a lysine side chain within its central pore. Lacking the insert in its third blade, the WDR34 β-propeller allows the associated copy of DYNC2H1 to adopt a bent conformation. The WDR34 and WDR60 β-propellers are held together by a tower of light chains, the base of which is a DYNLT1-DYNLT2B heterodimer, which we demonstrate using X-ray crystallography and SEC-MALS. The N-proximal region of WDR60 wraps around the DYNLT1-DYNLT2B heterodimer, providing an explanation of why the binding site is large relative

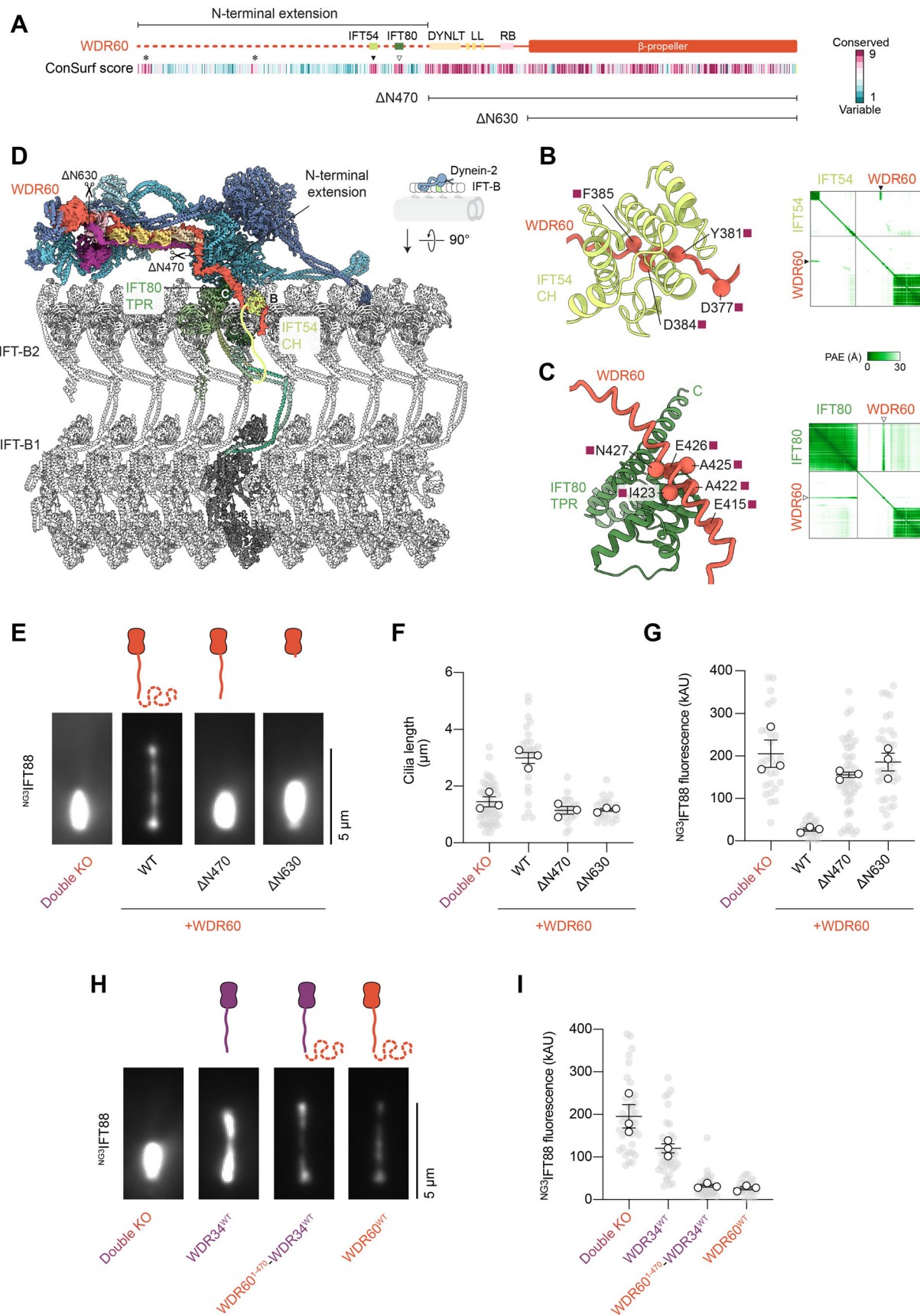

◀ **Figure 5. The N-terminal extension of WDR60 is a modular element that tethers dynein-2 to IFT-B.**

(A) Sequence diagram of WDR60, with features and protein-binding sites annotated. Below; the ConSurf score for each residue, colored according to the scale on the right. Variable residues (score 1) in cyan, conserved residues (score 9) in magenta. Conserved patches of residues in the N-terminal extension are marked with an asterisk or arrowhead. The extent of the WDR60$^{\Delta N470}$ and WDR60$^{\Delta N630}$ constructs is indicated below. (B) Left; AF Multimer model of the interface between WDR60 and the IFT54 CH domain. Conserved interface residues in WDR60 are indicated with colored squares according to the ConSurf score. Right; Predicted aligned error (PAE) plot. Note the high-confidence prediction (low PAE) for the interface between the IFT54 TPR domain and WDR60 residues 375–388 (black arrowhead). See Appendix Fig. S5A for the binding assay. (C) Left; AF Multimer model of the interface between WDR60 and the IFT80 TPR domain. Conserved interface residues in WDR60 are indicated with colored squares depicting the ConSurf score according to the scale in (A). Right; Predicted aligned error (PAE) plot. Note the high-confidence prediction (low PAE) for the interface between the IFT80 TPR domain and WDR60 residues 407–433 (white arrowhead). (D) Structural model of the dynein-2 complex bound to IFT-B in the anterograde IFT train, with the interfaces in (B) and (C) indicated, as well as truncation sites in WDR60$^{\Delta N470}$ and WDR60$^{\Delta N630}$. IFT-B coordinates from PDB 8BD7 (Lacey et al, 2023). The IFT-B2 and IFT-B1 subcomplexes are indicated. (E) Representative images showing time-averaged $^{NG3}$IFT88 signal in double KO cells expressing the indicated WDR60 constructs. Scale bar 5 μm. (F) Quantification of cilia length in double KO cells expressing the indicated WDR60 constructs from three technical replicates. Gray circles; individual data points. White circles; average from each separate experiment. Lines; mean (±SEM). 49 double KO, 27 WDR60$^{WT}$, 22 WDR60$^{\Delta N470}$, and 22 WDR60$^{\Delta N630}$ cilia were measured. Expression of WDR60$^{WT}$ restored cilia length (one-way ANOVA followed by Kruskal–Wallis test, $p < 0.0001$). WDR60$^{\Delta N470}$ and WDR60$^{\Delta N630}$ did not rescue cilia length ($p > 0.05$). (G) Quantification of $^{NG3}$IFT88 fluorescence intensity in double KO cells expressing the indicated WDR60 constructs from three technical replicates. Lines; mean (±SEM). 28 double KO, 28 WDR60$^{WT}$, 49 WDR60$^{\Delta N470}$, and 35 WDR60$^{\Delta N630}$ cilia were measured. Expression of WDR60$^{WT}$ rescued $^{NG3}$IFT88 accumulation in cilia (one-way ANOVA followed by Kruskal–Wallis test, $p < 0.0001$). WDR60$^{\Delta N470}$ and WDR60$^{\Delta N630}$ did not rescue $^{NG3}$IFT88 accumulation ($p > 0.05$). (H) Representative images showing time-averaged $^{NG3}$IFT88 signal in double KO cells expressing the indicated constructs. Scale bar 5 μm. (I) Quantification of $^{NG3}$IFT88 fluorescence intensity in double KO cells expressing the indicated constructs from three technical replicates. Lines; mean (±SEM). 39 double KO, 38 WDR34$^{WT}$, 41 WDR60$^{1-470}$-WDR34$^{WT}$ cilia were analyzed. WDR60$^{WT}$ data from 5G is shown for comparison. Expression of WDR34$^{WT}$ in double KO reduced $^{NG3}$IFT88 accumulation (one-way ANOVA followed by Kruskal–Wallis test, $p < 0.05$). WDR60$^{1-470}$-WDR34$^{WT}$ rescued $^{NG3}$IFT88 accumulation to the same extent as WDR60$^{WT}$ ($p < 0.001$ vs double KO; $p > 0.05$ vs WDR60$^{WT}$). Source data are available online for this figure.

to other DYNLT-binding proteins that interact with one face only (Hamada et al, 2018). N-terminal to the DYNLT1-DYNLT2B binding site WDR60 is a 470-residue flexible extension.

Our cell biology data and previous studies organize these interactions into a functional hierarchy (Fig. 6A–C). CRISPR-Cas9 knockouts show that cilia can assemble in the absence of WDR34 or WDR60 individually, though the loss of WDR60 causes a pronounced accumulation of cargoes, consistent with previous data (Vuolo et al, 2018; Hamada et al, 2018; Tsurumi et al, 2019; Shak et al, 2023; Hiyamizu et al, 2023b). Congruently, we show that retrograde IFT can persist in the absence of WDR34 or WDR60, albeit with impaired velocity and frequency. However, when both WDR34 and WDR60 are absent, cilia assembly fails and retrograde IFT is ablated. This implies that, minimally, at least one IC must engage DYNC2H1 for an appreciable level of cellular function. We interpret this finding in view of the following observations (1) the asymmetry of dynein-2 matches the shape of the IFT-B polymer in the anterograde train (Jordan et al, 2018; Toropova et al, 2019; Lacey et al, 2023); (2) WDR34 and WDR60 have not been found to homodimerize (Asante et al, 2014; Hamada et al, 2018); (3) WDR34 does not make direct contact with the anterograde IFT train, but the associated bent copy of DYNC2H1 does (Toropova et al, 2019; Lacey et al, 2023). Our data suggest that, in the case of WDR34, the binding of the IC and its associated light chains (Tsurumi et al, 2019) can partially stabilize the asymmetry of DYNC2H1, sufficient for a minimal level of attachment to anterograde IFT trains and transport into cilia in the absence of WDR60. This minimal amount of ciliary dynein-2, which is associated with a substantially reduced level of retrograde IFT, can support ciliogenesis but not the proper distribution of cargoes.

WDR60 has a functionally important role contributed by its long N-terminal extension (Fig. 6A). This region can be transplanted onto WDR34 and retain function, suggesting it serves as a modular flexible tether. Our structural model shows that the WDR60 N-terminal extension is ideally positioned to interact with the IFT80 TPR domain and IFT54 CH domain via two conserved patches of amino acids. There are a further two patches of amino

acids near the N-terminus of WDR60 that are conserved among vertebrates, which may serve as additional tether points, consistent with multivalent attachment of dynein-2 to IFT-B (Hiyamizu et al, 2023b). Multivalency would explain why the removal of the IFT54 CH domain has a mild effect on cilia function, as the multiple interactions made by the WDR60 N-terminal extension are likely to be partially redundant with each other (Hiyamizu et al, 2023a). In contrast, the IFT80 TPR domain is critical for ciliogenesis (Taschner et al, 2018). However, this domain also mediates dimerization with IFT172 in the IFT-B complex (Petriman et al, 2022; Lacey et al, 2023; Hesketh et al, 2022). The IFT80:IFT172 interface is non-overlapping with the putative WDR60 interface, suggesting that both interactions could occur simultaneously.

Our data do not exclude the loss of WDR60 or WDR34 having an effect on dynein-2 assembly. Indeed, our previous studies provide evidence for such a role (Vuolo et al, 2018; Toropova et al, 2019). Here, our experiments demonstrate that appending the N-terminal extension of WDR60 onto WDR34 enables remarkably full levels of dynein-2 function in the absence of WDR60. Similarly, deletion of WDR34 has only a mild effect on dynein-2 cellular function. Thus, dynein-2 is resilient to the loss of either the WDR34 or WDR60 β-propeller domain and light-chain binding sites, provided at least one IC with the WDR60 N-terminal extension is present. These results indicate that the contributions of the WDR34 or WDR60 β-propeller domain and light-chain binding sites to dynein-2 assembly are secondary to the role of the flexible WDR60 N-terminal region in targeting dynein-2 to cilia.

Why might dynein-2 use a flexible tether to help engage the IFT train? In the mature anterograde IFT train, dynein-2 complexes form a dense interdigitated array on the side of the train (Jordan et al, 2018; Toropova et al, 2019; Lacey et al, 2023). The large size and dense packing of dynein-2 suggest a considerable kinetic barrier would exist for each new dynein-2 complex to associate productively. We suggest that, by being flexibly tethered to the IFT train via the WDR60 N-terminal extension, dynein-2 complexes could settle into an optimal packing arrangement on the train (Fig. 6). A flexible tether between WDR60 and IFT-B might also

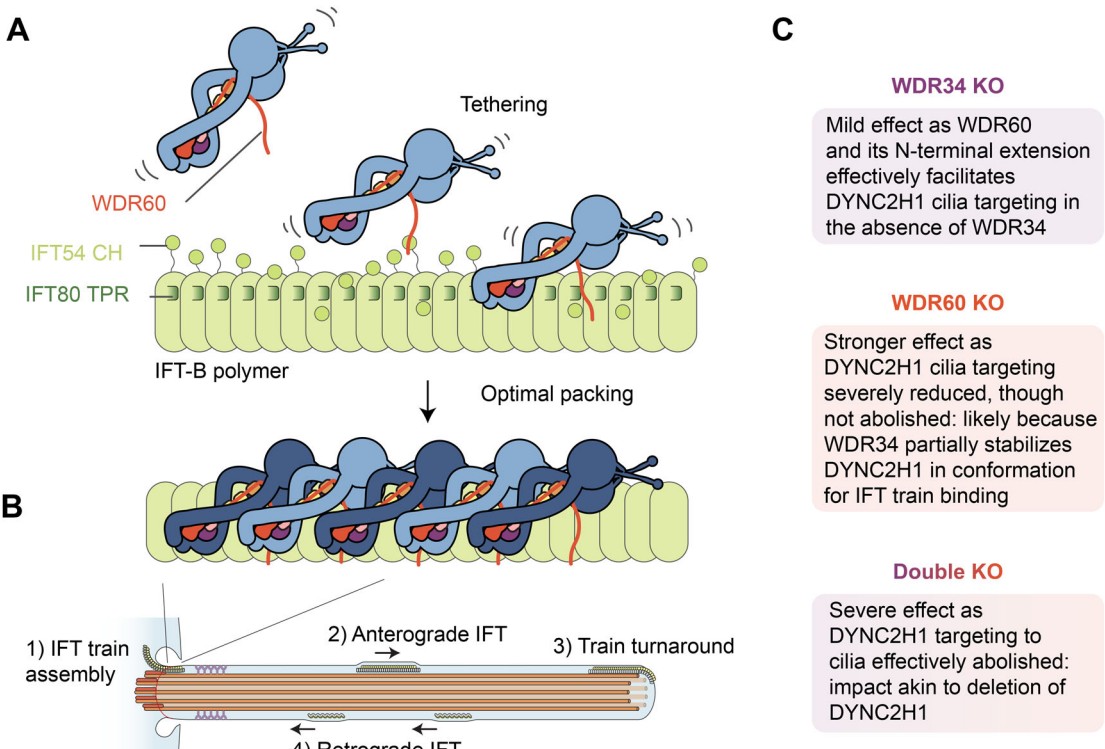

**Figure 6.  Model for dynein-2 tethering and attachment to anterograde IFT trains.**

(A) Schematic depiction of how the N-terminal extension of WDR60 could facilitate the initial encounter between dynein-2 and the anterograde IFT train assembling at the ciliary base, leading to optimal packing of dynein-2 complexes in the assembled polymer (B). Consequences of intermediate chain knockouts are summarized in panel C. See main text for details.

enable dynein-2 to remain loosely attached to IFT-B while it remodels from the anterograde IFT train into the retrograde train (Wingfield et al, 2021). Interestingly, the IFT-A complex also forms an intricate interwoven polymer in the context of the anterograde IFT train and uses long flexible tethers to engage IFT-B—in this case, contributed by the C-terminal regions of IFT88 and IFT172 (Kobayashi et al, 2021; Hesketh et al, 2022; Lacey et al, 2023). Similarly, tubulin also uses a disordered region of IFT74, in conjunction with the CH domain of IFT81, to bind to IFT-B (Bhogaraju et al, 2013; Kubo et al, 2016). Finally, the disordered linker between the CH and coiled-coil domain of IFT54 also serves a critical although not structurally understood role in retrograde IFT and regulating kinesin-2 (Zhu et al, 2021; Hiyamizu et al, 2023a). Thus, the use of flexible tethers may represent a general strategy for cargoes to dock to the assembling IFT train prior to attaining their intricate arrangement in the mature polymer.

## Methods

### Expression and purification of the dynein-2 complex

A plasmid for baculovirus-mediated expression of the dynein-2 heavy chain (DYNC2H1), two intermediate chains (WDR60/DYNC2I1 and WDR34/DYNC2I2), light intermediate chain (DYNC2LI1), and single isoforms of the four dynein-2 light chains

(DYNLRB1, DYNLL1, DYNLT1, and DYNLT2B/TCTEX1D2) was generated by removing redundant light chain isoforms (DYNLRB2, DYNLL2, DYNLT3, and LC8-like) from Addgene plasmid #132536 (Toropova et al, 2019) using restriction enzymes and HiFi assembly (NEB). The dynein-2 complex was purified from sf9 insect cells as described (Toropova et al, 2017, 2019).

### Expression, purification, and biophysical analysis of DYNLT1-DYNLT2B

Genes encoding human DYNLT1 and DYNLT2B were synthesized with codon optimization for *E. coli* (Eurofins Genomics) and inserted into pRSFDuet-1 (Novagen) with an N-terminal His8-TEV tag (DYNLT1) and N-terminal TwinStrepII-TEV tag (DYNLT2B) using HiFi assembly. Co-expression of DYNLT1 and DYNLT2B was induced in *E. coli* BL21 (DE3) at 20 °C for 16 h with 0.5 mM IPTG. Cells were harvested by centrifugation, washed in PBS, and then flash-frozen in liquid nitrogen prior to storage at −80 °C.

Protein purification was performed at 4 °C or on ice. Cell pellets were resuspended in Buffer A (100 mM Tris HCl pH 8, 150 mM NaCl, and 1 mM EDTA). Suspensions were sonicated at 60 Hz for 4 min with 30 s bursts and 30 s rest periods. Lysate was clarified by ultracentrifugation at 40,000 rpm for 30 min using a Beckman Optima centrifuge and a Type 70 Ti rotor. The supernatants were transferred to an IBA Strep-Tactin® XT Superflow high-capacity

gravity flow column pre-equilibrated with Buffer A. The column was washed five times with Buffer A. Three elutions were performed with Buffer A supplemented with 50 mM biotin. Size-exclusion chromatography with multi-angle light scattering (SEC-MALS) was performed using an Agilent 1100 instrument with a Superdex 200 10/300 column connected to a Wyatt Technologies Dawn 8+ and a Wyatt Technologies Optilab T-rex. Data were analyzed using Wyatt Technologies Astra 6.1.

## X-ray crystallography

For crystallography, DYNLT1-DYNLT2B was expressed as above, except a Sumo tag was added to each construct upstream of the TEV cleavage site to improve the yield. The first step of purification used IBA Strep-Tactin® XT Superflow high-capacity resin as described above, except the protein was released from the resin by incubating with TEV protease at 4 °C on a roller overnight (leaving the cleaved TwinStrepII-Sumo tag bound to the resin). To remove the cleaved His8-Sumo tag and His-tagged TEV protease, the solution was supplemented with NaCl to a final concentration of 300 mM and imidazole to a final concentration of 20 mM, prior to passing over HisPur Ni-NTA resin. DYNLT1-DYNLT2B was concentrated to 15 mg/ml using an Amicon® Ultra-15 Centrifugal Filter Unit with a 10 kDa molecular weight cut-off.

DYNLT1-DYNLT2B was crystallized using the hanging-drop vapor diffusion method by mixing 1 µl of protein solution (15 mg/ml) with 1–2 µl of precipitant (1.2–1.4 M potassium sodium tartrate, Tris pH 8–8.5) and incubating at 20 °C. Crystals were cryo-protected in mother liquor supplemented with 25–30% glycerol and then flash-frozen in liquid nitrogen. X-ray diffraction data were collected at the European Synchrotron Radiation Facility (ESRF; Grenoble, France) at beamline ID23-1 and indexed, processed, and scaled using the XDS suite (Kabsch, 2010). The reflection set was converted to binary using CCP4cloud (Winn et al, 2011). Initial phasing by molecular replacement in MOLREP (Vagin and Teplyakov, 1997) used the crystal structure of the *Drosophila* TCTEX-1 homodimer (PDB 1YGT) (Williams et al, 2005). The structure was obtained by iterative cycles of model building in Coot (Emsley et al, 2010) and ISOLDE (Croll, 2018) and refinement in PHENIX (Afonine et al, 2012) aided by a homology model of human DYNLT1 and DYNLT2B generated in Swiss-Model (Waterhouse et al, 2018). Data and refinement statistics are given in Appendix Table S3.

## Cryo-electron microscopy

### Sample vitrification
Dynein-2 was prepared for cryo-EM as described (Toropova et al, 2019), except a single 30 s application of the sample to the EM grid was used, and blot conditions were 6 s with a force setting of −10.

### Data collection
Data were collected on an in-house Titan Krios instrument (Thermo Fisher Scientific), 300 keV, equipped with a BioQuantum K3 direct electron detector and post-column GIF energy filter (slit width; 20 eV) (Gatan, Inc.). EPU software in "faster acquisition" mode was used for image collection with aberration-free image shift (AFIS). Images were collected at 105,000X nominal magnification as super-resolution, 50-frame movies (0.414 Å/pixel sampling). A 2.5 s exposure was used giving an electron exposure of 50.6 e⁻/Å². 

In total 12,975 micrographs were collected using a nominal defocus range of −1.5 to −3.5 µm.

### Image pre-processing
Pre-processing was carried out using cryoSPARC v3.2 (Punjani et al, 2017). Movies were aligned, dose-weighted, and summed using patch motion-correction. An output F-crop factor of 1/2 was used, which gave a 0.848 Å/pixel sampling for aligned and summed micrographs. CTF parameters were determined using patch CTF estimation with default settings.

### Intermediate chain reconstructions
Image processing was carried out using cryoSPARC v3.2. Five class averages of dynein-2 "tail" particles from Toropova et al 2019 data were used as templates for particle picking, low-pass filtered to 20 Å. Picks were pruned to exclude those with thick carbon background or high-intensity regions and extracted into 144-pixel boxes (3.312 Å/pixel sampling) giving a total of 2,286,627 putative particles. Ab initio reconstruction was used to generate initial 3D models from the data followed by multiple rounds of heterogeneous 3D refinement and 2D classification to select tail particles of dynein-2 containing the WDR34 and WDR60 intermediate chain subunits. Non-uniform 3D refinement was then used to obtain a consensus map from 113,479 such particles (extracted into 578 boxes, sampling 0.828 Å/pixel) at a global resolution of 3.9 Å. Local refinements were used to determine maps of WDR34 and WDR60 with improved local density (Appendix Fig. S1). LocScale (Jakobi et al, 2017) was used to locally sharpen the focused refinements.

### Model building
Models were generated of WDR60:DYNC2H1:DYNLRB1:DYN-C2LI1 and WDR34:DYNC2H1:DYNLRB1:DYNLL1 using Alpha-Fold2 Multimer (Evans et al, 2021; Jumper et al, 2021), fit into the locally refined maps of WDR60 and WDR34 respectively using UCSF Chimera (Pettersen et al, 2004), and used as starting points for molecular dynamics flexible fitting in ISOLDE using adaptive distance restraints (Croll, 2018; Croll and Read, 2021). Data and refinement statistics are given in Appendix Tables S1–2.

### AlphaFold analysis
AlphaFold2 Multimer predictions (Evans et al, 2021; Jumper et al, 2021) were run using ColabFold without the use of templates (Mirdita et al, 2022).

### Visualization
Cryo-EM maps and atomic coordinates were visualized using UCSF Chimera X (Pettersen et al, 2021).

## Mammalian cell culture and constructs

Mouse IMCD-3-FlpIn cell line (gift from P.K. Jackson, Stanford University) (Mukhopadhyay et al, 2010) and [NG3]IFT88 IMCD-3 cell line (gift from M. Nachury, UCSF) (Liew et al, 2014) were cultured in DMEM/F12 (11330-057; Gibco) supplemented with 10% FBS, 100 U/ml penicillin-streptomycin. Parental cell lines were verified to be free from mycoplasma (Sigma-Aldrich, MP0035). Cells were incubated in serum-free medium for 24 h to induce ciliogenesis. For Smo experiments, serum-starved cells were treated with SAG (Sigma-Aldrich, 566660) at a final concentration of 400 nM.

Genes encoding human WDR34 and WDR60 were synthesized by Epoch (Toropova et al, 2019). An mScarlet tag was added by PCR and Gibson assembly. For stable expression of mScarlet-tagged DYNC2H1, WDR34, WDR60, and other constructs in [NG3]IFT88 IMCD-3 cells using the Super PiggyBac system (System Biosciences), the relevant open reading frames were inserted into the PB-EF1α-MCS-IRES-Neo vector with N-terminal mScarlet tags TEV, FLAG, and PreScission Protease tags.

## Genome editing and verification

CRISPR-Cas9 genome editing experiments in IMCD-3 cells were performed as described (Hesketh et al, 2022; Perretta-Tejedor et al, 2020), with minor modifications. Guide RNAs (gRNAs) targeting DYNC2H1, WDR34 and WDR60 in IMCD-3 cells were designed using Benchling software and cloned into the pX330 vector (gift from Feng Zhang; Addgene plasmid # 42230) (Cong et al, 2013):

5'-TCCGGGCAGTTGCTCAACGG-3' targeting exon 1 of DYNC2H1

5'-TGGCAATGTGCGCGTCGCCC-3' targeting exon 1 of WDR34

5'-GCAGTCAGGCAGTCCCAAGG-3' targeting exon 3 of WDR60

Transfection of Cas9 vectors expressing relevant gRNA was performed as described previously (Hesketh et al, 2022; Perretta-Tejedor et al, 2020). In a well of a six-well plate, 50–60% confluent IMCD-3 cells were transfected with 2.4 µg plasmid DNA, using 8.5 µl Lipofectamine 2000 transfection reagent (Thermo Fisher Scientific) in 300 µl Opti-MEM reduced serum medium (Thermo Fisher Scientific). About 2.4 µg plasmid DNA was made up in equal parts of each expression vector and mScarlet vector (i.e., 1.2 µg each vector for co-transfections). For positive control of transfection, cells were transfected with the mScarlet expression vector alone (1.2 µg as a control for co-transfections). For negative control, cells were mock-transfected with transfection reagent in Opti-MEM, but no plasmid DNA. mScarlet+ cells were selected by FACS. A population of mScarlet+ cells was collected from the group sort. mScarlet+ cells from group sort were cultured for two weeks before single cell sorting by FACS. Single-cell sorted cells were seeded in 96 well plates in a 150 µl conditioned medium. The conditioned medium was prepared by sterile filtration of a 1:1 mixture of fresh medium, and medium removed from a flask of cells in the exponential growth phase (50–70% confluent). Single-cell sorted colonies were expanded by a subculture to increasingly larger culture vessels (24 well plates, 12-well plates, and T25 flask).

To confirm for mutations, genomic DNA was isolated (Lucigen, QE0905T) and targeted sequences were PCR amplified. PCR products were cloned into pJET1.2/blunt vector (Thermo Fisher, K1232) and subsequently sequenced. Clones with insertions/deletions leading to frameshift mutations eventually with a premature STOP codon were considered as KO candidates. A double KO cell line was generated by knocking out WDR34 in the confirmed WDR60 KO cell line.

## Immunoblotting

Cells were lysed in RIPA buffer containing 150 mM sodium chloride, 1.0% Triton X-100, 0.5% sodium deoxycholate, 0.1% SDS, 50 mM Tris, pH 8.0. Samples were separated by SDS-PAGE followed by transfer to nitrocellulose membranes. For anti-DYNC2H1 blots, membranes were blocked in 5% skimmed milk-TBST. Anti-DYNC2H1 (gift from Prof. Richard Vallee; 1:150) and anti-GAPDH (1:10,000; Cell Signaling Technology, 2118) diluted in blocking buffer were incubated with membrane overnight and detected using HRP-conjugated secondary antibodies (Jackson ImmunoResearch) and enhanced chemiluminescence (GE Healthcare). For anti-FLAG blots, membranes were blocked overnight in 3% milk-TBST and incubated for 4 hr in ANTI-FLAG® M2 mouse antibody (1:1000; Sigma-Aldrich, F1804) and anti-GAPDH (1:5000; Cell Signaling Technology, 2118).

## Immunofluorescence

IMCD-3 cells grown on 0.17 mm thick (#1.5) cover glass (VWR, 631-0150 P) in 12-well plates were first washed with PBS, followed by two washes in cytoskeletal buffer (100 mM NaCl, 300 mM sucrose, 3 mM MgCl$_2$, 10 mM PIPES) and fixed in 4% paraformaldehyde prepared in cytoskeletal buffer with 0.5% Triton and 5 mM EGTA as described previously (Hesketh et al, 2022). Next, cells were blocked in 3% BSA and 2% FBS in PBS for 1 h at room temperature. Cells were incubated overnight with respective antibodies. Post-primary antibody incubation, cells were washed and incubated with corresponding secondary antibody in a blocking solution for 1 h. Coverslips were mounted onto glass slides using mounting medium + DAPI [4,6-diamidino-2-phenylindole (GeneTex, GTX30920)] for nuclear staining. Cells were imaged by TIRF microscopy. The following antibodies were used at indicated dilutions: anti-acetylated tubulin (Sigma-Aldrich T6793; 1:2000); anti-gamma-tubulin (Sigma-Aldrich T6557; 1:500); anti-Smo (Santa Cruz sc-166685; 1:500); anti-DYNC2H1 (ABCAM ab225946; 1:100). AlexaFluor-labeled secondary antibodies (Thermo Fisher) were used at 1:500 dilution.

## Stable cell line construction

To stably express mScarlet-tagged proteins in [NG3]IFT88 IMCD-3 cells, we used the Super PiggyBac transposon vector system (System Biosciences). Cells in six-well plates were co-transfected with PiggyBac plasmid containing mScarlet-tagged gene of interest (DYNC2H1, WDR34, WDR60, WDR60$^{\Delta N470}$, WDR60$^{\Delta N630}$, and WDR60$^{N470}$-WDR34$^{WT}$) and a geneticin resistance marker (0.5 µg) and 0.2 µg of Super PiggyBac transposase expression vector using Lipofectamine 2000. Clones were selected using geneticin resistance (500 µg/ml) after 2 d of transfection until confluent and screened by live-cell TIRF microscopy to confirm the expression of both mNeonGreen and mScarlet labeled proteins. For continued culture, growth media containing 500 µg/ml geneticin and 100 µg/ml hygromycin was used.

## TIRF microscopy

All imaging was conducted on an Eclipse Ti-E inverted microscope with a CFI Apo TIRF 1.49 NA oil objective, Perfect Focus System, H-TIRF module, LU-N4 laser unit (Nikon), and a quad-band filter set (Chroma). Images were captured on an iXon DU888 Ultra EMCCD camera (Andor), controlled with NIS-Elements AR Software (Nikon). The microscope was kept in a temperature-controlled environmental chamber (Okolab). Files were imported into Fiji (ImageJ, NIH) (Schindelin et al, 2012) and analyzed. Kymographs were generated using KymographClear (Mangeol et al, 2016).

### Live-cell imaging

Approximately 150,000 cells were seeded onto a 35 mm high glass bottom dish (Thistle Scientific, 81158) and grown for 16–24 h. Subsequently, cells were starved for 24 h in serum-free media and transferred for live-cell imaging by TIRF microscopy with 100-ms exposures at 37 °C. Cells were imaged in DMEM/F12, HEPES with no phenol red (Gibco, 11039) for not more than 1 h per dish.

### Pull-down experiment

Human ZZ-tagged IFT54$^{CH}$ (IFT54 residues 1-137), ZZ-SNAP-GST-WDR60$^{355-449}$ or ZZ-SNAP-GST were cloned and expressed using the baculovirus system as previously described (Toropova et al, 2019; Hesketh et al, 2022). Cells were harvested by centrifugation, washed in PBS, then flash-frozen in liquid nitrogen prior to storage at −70 °C.

IFT54$^{CH}$ was purified at 4 °C or on ice. Cell pellets were resuspended in lysis buffer (30 mM HEPES pH 7.4, 150 mM KCl, 1 mM EGTA, 1 mM DTT, 1 mM PMSF, 10% v/v glycerol, cOmplete EDTA-free Protease Inhibitor Cocktail (Roche)). Cells were lysed using a Dounce homogenizer with 10 strokes with a tight clearance pestle. Lysates were clarified via ultracentrifugation in a Type 70 Ti rotor at 184,800 × *g* for 30 min. The supernatant was incubated for 1 h on a roller with 500 µl IgG Sepharose 6 resin (GE Healthcare) that had been pre-washed in wash buffer (lysis buffer lacking protease inhibitor cocktail). The resin was then transferred to a 10 ml poly-prep chromatography column (BioRad), washed with 2 × 10 ml volumes of wash buffer and 1 × 5 ml volume of TEV buffer (50 mM Tris pH 7.5, 150 mM K-acetate, 2 mM Mg-acetate, 1 mM EGTA, 10% glycerol, and 1 mM DTT) and transferred to a 2-ml microfuge tube (1 ml total volume). The resin suspension was supplemented with 100 µg TEV protease and incubated overnight with gentle rolling. TEV-cleaved proteins were separated from the resin using an empty column. Aliquots of 100 µl were flash-frozen in liquid nitrogen and stored at −70 °C.

For the pull-down assay, ZZ-SNAP-GST-WDR60$^{355-449}$ or ZZ-SNAP-GST were immobilized on 80 µl IgG Sepharose 6 resin as described for the initial steps of the ZZ-IFT54$^{CH}$ purification. The resin was washed with 2 × 500 µl wash buffer. Purified IFT54$^{CH}$ was diluted to 4.6 µM in wash buffer and 500 µl of diluted IFT54$^{CH}$ solution was flowed over the resin bearing either SNAP-GST-WDR60$^{355-449}$ or SNAP-GST followed by 2 × 500 µl washes with wash buffer. The resin was then transferred to a 2-ml microfuge tube (160 µl volume). The resin suspension was supplemented with TEV protease and incubated overnight with gentle rolling. TEV-cleaved proteins were separated from the resin by centrifugation and 12 µl of eluates were analyzed by SDS-PAGE on NuPAGE 4–12% Bis-Tris gels in MES running buffer with Sypro Red staining (Thermo Fisher Scientific).

### Statistical analyses

Statistical analyses were performed with GraphPad Prism v8.3.0. Pairwise comparisons were made using Mann–Whitney tests and multiple comparisons were made using one-way ANOVA followed by Kruskal–Wallis tests, which do not assume normal distributions. Superplots were generated as described previously (Lord et al, 2020).

### Data availability

The datasets produced in this study are available in the following databases:

X-ray crystallography data (DYNLT1-DYNLT2B): Protein Data Bank accession code 8RGI (https://doi.org/10.2210/pdb8rgi/pdb)

Cryo-electron microscopy data (WDR34): Electron Microscopy Data Bank accession code EMD-19132 (https://www.ebi.ac.uk/emdb/EMD-19132) and Protein Data Bank accession code 8RGG (https://doi.org/10.2210/pdb8RGG/pdb)

Cryo-electron microscopy data (WDR60): Electron Microscopy Data Bank accession code EMD-19133 (https://www.ebi.ac.uk/emdb/EMD-19133) and Protein Data Bank accession code 8RGH (https://doi.org/10.2210/pdb8RGH/pdb).

### Peer review information

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

## Acknowledgements

We thank Peter Jackson (Stanford) and Max Nachury (UCSF) for cell lines; Saúl Álvarez-Teijeiro (Birkbeck) for assistance with Western blotting; Great Ormond Street Institute of Child Health and Queen Mary University of London

for cell sorting facilities; Richard Vallee (Columbia University) for anti-DYNC2H1 antibody; Caroline Shak, Borhan Uddin, and Johannes Weijman (University of Bristol) for helpful input into the project; Natasha Lukoyanova and Shu Chen (Birkbeck) for cryo-EM support; Nikos Pinotsis (Birkbeck) for X-ray crystallography support; and David Houldershaw, Yanni Goudetsidis, and Richard Westlake (Birkbeck) for computational support. This work is funded by grants from UKRI-BBSRC (BB/S013024/1, BB/S005390/1, BB/P008348/1, and BB/S007202/1); the Wellcome Trust (217186/Z/19/Z and 210585/Z/18/Z); and the Royal Society (RG170260). This research was also supported by the National Institute for Health Research Biomedical Research Centre at Great Ormond Street Hospital for Children NHS Foundation Trust and University College London. Cryo-EM data for this investigation were collected at Birkbeck College, University of London with financial support from the Wellcome Trust (202679/Z/16/Z and 206166/Z/17/Z). X-ray crystallography and biophysical data were collected at the biophysX facility at Birkbeck College, University of London with financial support from the Wellcome Trust (221543/Z/20/Z). The X-ray crystallography dataset was collected at beamline ID23-1 of the European Synchrotron Radiation Facility (ESRF) with the support of Gordon Leonard.

## Author contributions

**Aakash G Mukhopadhyay**: Investigation; Methodology; Writing—original draft; Writing—review and editing. **Katerina Toropova**: Investigation; Methodology; Writing—original draft; Writing—review and editing. **Lydia Daly**: Investigation; Methodology; Writing—review and editing. **Jennifer N Wells**: Investigation; Methodology; Writing—review and editing. **Laura Vuolo**: Investigation; Methodology; Writing—review and editing. **Miroslav Mladenov**: Investigation; Writing—review and editing. **Marian Seda**: Methodology; Writing—review and editing. **Dagan Jenkins**: Supervision; Funding acquisition; Methodology; Writing—review and editing. **David J Stephens**: Conceptualization; Supervision; Funding acquisition; Methodology; Writing—review and editing. **Anthony J Roberts**: Conceptualization; Supervision; Funding acquisition; Investigation; Methodology; Writing—original draft; Writing—review and editing.

## Disclosure and competing interests statement

The authors declare no competing interests.

