## [Peer Review File · The EMBO Journal]

Structure and Tethering Mechanism of Dynein-2 Intermediate Chains in Intraflagellar Transport

Aakash Mukhopadhyay, Katerina Toropova, Lydia Daly, Jennifer Wells, Laura Vuolo, Miroslav Mladenov, Marian Seda, Dagan Jenkins, David Stephens, and Anthony Roberts

Corresponding author: Anthony Roberts (anthony.roberts@path.ox.ac.uk)

Review Timeline:

Submission Date:	5th Sep 23
Editorial Decision:	4th Oct 23
Revision Received:	24th Dec 23
Editorial Decision:	15th Jan 24
Revision Received:	6th Feb 24
Accepted:	9th Feb 24

Editor: Ieva Gailite

Transaction Report:

Dear Anthony,

Thank you for submitting your manuscript for consideration by the EMBO Journal. We have now received comments from four reviewers, which are included below for your information.

As you will see from the reports, all reviewers find the study of interest and appreciate the quality of the data, while also pointing out a number of aspects that would need to be strengthened in the final manuscript. Based on the interest expressed in the reports, I would like to invite you to address the issues raised by the referees in a revised manuscript. I think it would be useful to discuss the revision in more detail via email or phone/videoconferencing - please let me know which option you prefer.

We generally allow three months as standard revision time, which can be extended to six months in the case of major revisions. Should you foresee a problem in meeting the three-month deadline, please let us know in advance to discuss an extension.

As a matter of policy, competing manuscripts published during this period will not negatively impact on our assessment of the conceptual advance presented by your study. However, please contact me as soon as possible upon publication of any related work to discuss the appropriate course of action.

When preparing your letter of response to the referees' comments, please bear in mind that this will form part of the Review Process File and will therefore be available online to the community. For more details on our Transparent Editorial Process, please visit our website: <https://www.embopress.org/page/journal/14602075/authorguide#transparentprocess>. Please also see the attached instructions for further guidelines on preparation of the revised manuscript.

Please feel free to contact me if you have any further questions regarding the revision. Thank you for the opportunity to consider your work for publication. I look forward to discussing your revision.

With best wishes,

Ieva

We realize that it is difficult to revise to a specific deadline. In the interest of protecting the conceptual advance provided by the work, we recommend a revision within 3 months (2nd Jan 2024). Please discuss the revision progress ahead of this time with the editor if you require more time to complete the revisions.

Referee #1:

The study presented in this paper significantly advances our understanding of Dynein-2 regulation and its interactions with intermediate chains WDR34 and WDR60. The combination of single particle cryo electron microscopy (cryo-EM) to determine structures of higher resolution than previously reported, crystallography, computational modeling and cellular experiments provides valuable insights into the molecular mechanisms underlying dynein-2 function. While some of the cell biology experiments represent re-investigations of previously published data (Vuolo, 2018), the findings presented in this manuscript do contribute to the field of cilia assembly and intracellular transport.

Major Concerns

In figure 5, the authors use alphafold to map the potential interactions of IFT-B proteins with WDR60 (Fig. 5B-C). These alphafold models are then used in the model of dynein bound anterograde IFT train in Fig. 5D to illustrate how WDR60 may mediate the interaction with the train. However, as the interacting parts are flexible, there is no cryo-EM density to back up the model. This is followed up by cell biology where it is shown the N-term deletion of WDR60, which is the part that interact with IFT-B proteins, does not rescue the cilogenesis phenotype of WDR34/60 double KO. In my opinion this is not sufficient evidence to support their model. The authors will need to provide additional biochemical and/or structural biology experiments to address the direct WDR60-IFT-B interactions. It would also be important to exclude the possibility that the KO of WDR60, while not affecting HC stability, could influence dynein-2 assembly and subsequently impact HC ciliary localization. This is particularly important given the observation that the truncated WDR60 Q631* mutant exhibited lower binding ability to IFT complexes than WDR60 WT, as reported in Vuolo, 2018, which may suggest that there are other regions of WDR60 important for IFT-train association and at least warrants discussion.

ADDITIONAL MINOR CONCERNS:

1. In Figure 5, the authors assert that WDR60 tethers to the IFT trains via the IFT54 CH-domain and the very C-terminus of IFT80. However, it is unclear whether this C-terminal region of IFT80 is the same as the region previously examined in Taschner et al., 2018, which was shown to significantly reduce the percentage of ciliated IMCD3 cells. If this is indeed the same region, the authors may have identified the molecular basis for why the deletion of the C-terminal TPR region of IFT80 hampers cilogenesis. Discussion of this point is necessary.
2. On page 10, it should read "loss of WDR34 or WDR60," not "loss of WDR34 and WDR60," as the double knockout (KO) cells exhibited no detectable intraflagellar transport (IFT). Please correct this discrepancy for accuracy.
3. On page 13, the reference to Hiyamizu et al., 2023b, regarding the interaction between IFT80 and WDR60 should be clarified. It appears that the authors are referencing Figure 1F. However, the VIP assay in Figure 1E shows no binding between IFT80 and any dynein-2 subunit, and this evidence should thus not be used to support the predicted interaction.
4. The authors should also address the discrepancies between the findings reported in this study and those in Vuolo, 2018. Specifically, the differences in the role of WDR34 in cilia extension and the requirement for the C-terminal beta-propeller domain of WDR60 in binding IFT-B proteins should be discussed.
5. On page 4, there is a typographical error: "focussed" should be corrected to "focused."

Referee #2:

The research study by Aakash and colleagues is an exemplar of integrative approaches, employing cryo-EM, AlphaFold predictions, X-ray analysis, and CRISPR/Cas9-mediated cell biology techniques to dissect the complex regulation of dynein-2 by WDR34 and WDR60. Their insights reveal distinct interaction modalities between the dynein-2 intermediate chains and their respective heavy chains.

Interestingly, the study shows that neither WDR34 nor WDR60 alone is indispensable for cilia assembly; it is the concurrent absence of both that disrupts the process. Moreover, WDR60 is identified as playing a more prominent role in the dynein-2 machinery, with its N-terminal extension serving as a critical factor for ciliary targeting. The authors theorize that this extension acts as a flexible tether essential for the assembly of IFT trains at the ciliary base.

While the meticulous design and commendable results make this paper a significant contribution, especially in elucidating the role of unstructured tethers in IFT train interactions, several points require further clarification:

1. Assembly of Dynein-2 in Double Knockouts: The paper could be enhanced by addressing how dynein-2 assembles when both WDR34 and WDR60 are knocked out. Failure to do so could call into question some of the conclusions about double knockout effects and the role of the N-terminal tether. Negative staining analysis might be one way to investigate this.
2. Role of WDR60's N-terminal Extension in Ciliogenesis: The paper raises the question of why ciliogenesis is not abolished when WDR60 is knocked out, despite its significant role in the dynein-2 mechanism. Addressing this inconsistency would strengthen the paper's conclusions. Also, it might be very intriguing if this tether can be transplanted to other non-cilia proteins for ciliary targeting. It will be great if the authors can test it.
3. Validation of AlphaFold Predictions: Although AlphaFold is a powerful tool for protein structure prediction, it cannot replace the need for empirical evidence. Biochemical assays validating the key residues as indicated by AlphaFold would lend greater credence to the findings.
4. Statistics: The paper would benefit from clearly indicating the p-values on figures to allow for a robust statistical interpretation.

Referee #3:

Structure and tethering mechanism of Dynein-2's heterodimeric intermediate chains
Mukhopadhyay et al.

Dyneins are the main motors responsible for anterograde transport along microtubules and are far more complex than the other cytoskeletal motors, kinesins and myosins. In cilia and flagella, this anterograde movement is powered by dynein-2. Although dynein-2 is a homodimer of identical heavy chains, it is unusual in that it contains two different intermediate chains-WDR34 and WDR60-in contrast to its cytoplasmic counterpart, which contains two copies of the same polypeptide. Symmetry mismatches are always interesting in structural biology, and this one is even more so given the regulatory role played by the dynein-2 intermediate chains.

In this manuscript, Mukhopadhyay and colleagues use a combination of cryo-EM, crystallography, and cell-based functional assays to reveal how WDR34 and WDR60 interact with each copy of the dynein heavy chain in different ways and how the two intermediate chains have different functional roles. The most interesting result (in this reviewer's opinion) was that the long N-terminal extension of WDR60 is involved in the targeting of dynein-2 to cilia, and that its function could be preserved by transferring it to WDR34 in a cell lacking WDR60.

This is a very well written manuscript presenting an important contribution to our understanding of a complex motor with an even more complex regulation. The data will be of interest to anyone studying dynein, as well as to those studying cilia biology. I have no major concerns with the manuscript. I outline a couple of minor suggestions below that I believe could further strengthen it.

- Figure 1H-J. The authors state that while previous data suggested that DYNLT1 and DYNLT2B could form a heterodimer, a dimer of dimers could not be ruled out. The crystal structure presented in Fig.1I clearly shows that DYNLT1 and DYNLT2B can form a heterodimer. Would it be possible to dock this structure into the density shown in Fig.1H to rule out the possibility that a dimer of heterodimers can account for it? I can't tell whether the resolution in that region of the map isn't good enough to do this in an unambiguous manner, but if it is, it could put this (minor) issue to rest.
- While I agree that the most parsimonious conclusion for the data presented is that "the N-terminal extension of WDR60 is a modular element that can be transplanted onto WDR34 to rescue function", I am not 100% convinced that the data available out there rule out the possibility that one of the two WDR variants could occupy both binding sites if the other one is absent from the cell. If that were the case, then it would be the WD40s that are "modular", while the N-terminal extension of WDR60 would be found in its normal place.

Since I find this possibility far less likely than the one proposed by the authors, I would not expect them to do any additional experiments to address it. But I was wondering if something simple, such as a few runs with AlphaFold Multimer, could provide additional support for their proposal. For example, can AlphaFold Multimer come up with WDR34-WDR34 or WDR60-WDR60 homodimers that could still fit into the dynein-2 complex? I find the statement in the Discussion that "WDR34 and WDR60 cannot homodimerize" a bit too strong given the papers cited. I agree that those papers showed that WDR34 and WDR60 did

not dimerize under their conditions and with the approaches used, but "did not homodimerize" is quite different from "cannot homodimerize". Similarly, would it be possible to run AlphaFold Multimer using either two copies of WDR34 or of WDR60 plus the surrounding region of dynein's heavy chain? Even if absence of evidence (i.e. if AlphaFold cannot come up with structures in any of those cases) is not evidence of absence, this would help me further dismiss my alternative hypothesis. If none of these suggestions work, I would be satisfied if the Discussion acknowledged that the possibility that one WDR can occupy both binding sites, however unlikely, has not been completely ruled out.

Referee #4:

Mukhopadhyay et al. reported the structure and regulatory mechanisms of IFT-dynein by its two distinct intermediate chains WDR34 and WDR60. IFT-dynein powers retrograde IFT and its dysfunction causes many types of ciliary diseases; however, its regulatory mechanism remains unclear. Using cryo-EM and other structure biological approaches, this study shows that the WDR60 β -propeller uses a unique insert in its third blade to stabilize DYNC2H1 in a straight confirmation and that WDR34 uses a classic β -propeller mode interaction to engage DYNC2H1. They showed that the WDR34 and WDR60 β -propellers are held together by light chains. Their data suggest that the N-proximal region of WDR60 may act as a flexible tether to IFT-particles, providing additional insights for the ciliary entry and anterograde transport of IFT-dynein. Using Cas9-based knockout and imaging tools, this study revealed that WDR34 and WDR60 have redundant function in the formation of cilia. They showed that WDR60 appears to play a more important role likely because its N-terminal extension facilitates ciliary targeting of dynein-2. Many of the experiments are well performed. In particular, the visualization of dynein-2 heavy chain is not trivial but very informative. The manuscript is nicely written. While the characterization of detailed interaction of dynein-2's heavy chain, associated chains, and IFT-particles appears to be incremental, the idea of such a unique tethering mechanism is important and intriguing. I support its publication in EMBO J as long as the following minor issues can be addressed through discussion and rewriting.

1. The author can add one paragraph to discuss whether and how missense mutations of WDR34 and WDR60 are associated with any types of ciliopathies. Are there any mutations detected around the tethering interface?
2. How common is the redundancy of WDR34 and WDR60 in other ciliated organisms? Green algae or nematodes?
3. The model figure 6 can be expanded to illustrate the consequence of WDR34 or WDR60 KOs.

We are grateful to the Reviewers for their comments and suggestions, which we have addressed as detailed in the point-by-point response below.

Referee #1:

The study presented in this paper significantly advances our understanding of Dynein-2 regulation and its interactions with intermediate chains WDR34 and WDR60. The combination of single particle cryo electron microscopy (cryo-EM) to determine structures of higher resolution than previously reported, crystallography, computational modeling and cellular experiments provides valuable insights into the molecular mechanisms underlying dynein-2 function. While some of the cell biology experiments represent re-investigations of previously published data (Vuolo, 2018), the findings presented in this manuscript do contribute to the field of cilia assembly and intracellular transport.

Major Concerns

In figure 5, the authors use alphafold to map the potential interactions of IFT-B proteins with WDR60 (Fig. 5B-C). These alphafold models are then used in the model of dynein bound anterograde IFT train in Fig. 5D to illustrate how WDR60 may mediate the interaction with the train. However, as the interacting parts are flexible, there is no cryo-EM density to back up the model. This is followed up by cell biology where it is shown the N-term deletion of WDR60, which is the part that interact with IFT-B proteins, does not rescue the cilogenesis phenotype of WDR34/60 double KO. In my opinion this is not sufficient evidence to support their model. The authors will need to provide additional biochemical and/or structural biology experiments to address the direct WDR60-IFT-B interactions.

We have performed a pulldown assay with purified proteins to confirm the direct interaction between the N-terminal region of WDR60 (residues 355-449) and the CH domain of IFT54 in IFT-B (new Appendix Figure S5A). This interaction is also well supported by previous immunoprecipitation data (Hiyamizu *et al.*, 2023b). We also sought to perform a pulldown assay with IFT80, but the poor solubility of purified human IFT80 prevented this. Therefore, we have emphasized in the manuscript that the WDR60^{N-term}-IFT80 interaction remains putative (being based on AlphaFold prediction and previous immunoprecipitation and mass spectrometry data rather than a direct binding assay). In summary, our pulldown assay with purified WDR60³³⁵⁻⁴⁴⁹ and IFT54^{CH} provides biochemical evidence for direct WDR60:IFT-B interaction.

It would also be important to exclude the possibility that the KO of WDR60, while not affecting HC stability, could influence dynein-2 assembly and subsequently impact HC ciliary localization. This is particularly important given the observation that the truncated WDR60 Q631 mutant exhibited lower binding ability to IFT complexes than WDR60 WT, as reported in Vuolo, 2018, which may suggest that there are other regions of WDR60 important for IFT-train association and at least warrants discussion.*

We are grateful for this comment, which may have arisen from our lack of clarity in the original submission. We don't exclude that loss of WDR60 could have an effect on dynein-2 assembly. Indeed, our previous studies provide evidence for such a role (Vuolo *et al.*, 2018; Toropova *et al.*, 2019). Crucially, however, our experiments show that appending the N-terminal extension of WDR60 onto WDR34 enables remarkably full levels of dynein-2 function in the absence of WDR60. Thus, dynein-2 is strikingly resilient to the loss of WDR60's β -propeller domain and light-chain binding sites, provided at least one IC with the WDR60 N-terminal extension is present. These results indicate that any contributions of WDR60's β -propeller domain and light-chain binding sites to dynein-2 assembly are

dispensable and secondary to the role of the WDR60 N-extension region in targeting dynein-2 to cilia. We have added this clarification to the Discussion. We find that our model is fully consistent with WDR60 Q631* mutant analysis in Vuolo *et al.* (2018): the WDR60 Q631* mutant incorporates less efficiently into the dynein-2 complex, as it lacks the DYNC2H1-binding β -propeller domain; it therefore pulls down IFT-B proteins ineffectively compared to WT WDR60, as less DYNC2H1 (which binds directly to IFT172 and IFT80 in IFT-B) is present. These data are consistent with the view that full binding of dynein-2 to IFT-B requires the synergistic action of multiple weak interactions involving both DYNC2H1 and the N-terminal extension of WDR60.

ADDITIONAL MINOR CONCERNS:

*1. In Figure 5, the authors assert that WDR60 tethers to the IFT trains via the IFT54 CH-domain and the very C-terminus of IFT80. However, it is unclear whether this C-terminal region of IFT80 is the same as the region previously examined in Taschner *et al.*, 2018, which was shown to significantly reduce the percentage of ciliated IMCD3 cells. If this is indeed the same region, the authors may have identified the molecular basis for why the deletion of the C-terminal TPR region of IFT80 hampers ciliogenesis. Discussion of this point is necessary.*

We have addressed this point in the Discussion as suggested. In brief, the IFT80 site identified lies within the region previously deleted by Taschner *et al.* (2018), but the deleted region also includes an IFT172 binding site, so the cellular consequences of the deletion cannot be attributed to loss of dynein-2 interactions alone. The IFT80:IFT172 interface is non-overlapping with the putative IFT80:WDR60 interface, suggesting that both interactions could occur simultaneously.

2. On page 10, it should read "loss of WDR34 or WDR60," not "loss of WDR34 and WDR60," as the double knockout (KO) cells exhibited no detectable intraflagellar transport (IFT). Please correct this discrepancy for accuracy.

Amended. Thank you for improving the clarity of this text.

*3. On page 13, the reference to Hiyamizu *et al.*, 2023b, regarding the interaction between IFT80 and WDR60 should be clarified. It appears that the authors are referencing Figure 1F. However, the VIP assay in Figure 1E shows no binding between IFT80 and any dynein-2 subunit, and this evidence should thus not be used to support the predicted interaction.*

We have clarified this point in the text. In Hiyamizu *et al.* (2023b), co-immunoprecipitation of IFT80 with WDR60 is detected by mass spectrometry (Table 1) and western blotting (Figure 1F). The interaction is not perceptible in the visual immunoprecipitation (VIP) assay (Figure 1E). As Hiyamizu *et al.* write “*the expression levels and stability of individual proteins could vary from protein to protein and be affected by co-expressed proteins and that the interactions could be affected by the fluorescent protein tags. In addition, ‘not detected’ in the VIP assay does not necessarily mean ‘no interaction’*”. We think this is especially likely in the case of the IFT80 C-terminal domain, as there is evidence that this domain does not fold well when IFT80 is expressed on its own (based on the crystal structure and limited proteolysis analysis of Taschner *et al.* 2018).

4. The authors should also address the discrepancies between the findings reported in this study and those in Vuolo, 2018. Specifically, the differences in the role of WDR34 in cilia extension and the requirement for the C-terminal beta-propeller domain of WDR60 in binding IFT-B proteins should be discussed.

We recently published a study (Shak *et al.*, 2023) to confirm the basis for the WDR34 phenotype in Vuolo *et al.* (2018). In brief, the WDR34 mutant in Vuolo *et al.* (2018) retains expression of an N-terminal fragment, which exerts a dominant negative effect on cilia formation, in agreement with previous work (Tsurumi *et al.*, 2019). We highlight these papers and the dominant negative effect of the WDR34 N-terminal region in the Introduction. As mentioned in point 2 (and repeated here for clarity), we find our study to be consistent with the WDR60 data in Vuolo *et al.* (2018): the WDR60 Q631* mutant incorporates less efficiently into the dynein-2 complex, as it lacks the DYNC2H1-binding β -propeller domain; it therefore pulls down IFT-B proteins less effectively compared to WT WDR60, as less DYNC2H1 (which binds directly to IFT172 and IFT80 in IFT-B) would be present.

5. On page 4, there is a typographical error: "focussed" should be corrected to "focused."

Amended to the American spelling as suggested.

Referee #2:

The research study by Aakash and colleagues is an exemplar of integrative approaches, employing cryo-EM, AlphaFold predictions, X-ray analysis, and CRISPR/Cas9-mediated cell biology techniques to dissect the complex regulation of dynein-2 by WDR34 and WDR60. Their insights reveal distinct interaction modalities between the dynein-2 intermediate chains and their respective heavy chains.

Interestingly, the study shows that neither WDR34 nor WDR60 alone is indispensable for cilia assembly; it is the concurrent absence of both that disrupts the process. Moreover, WDR60 is identified as playing a more prominent role in the dynein-2 machinery, with its N-terminal extension serving as a critical factor for ciliary targeting. The authors theorize that this extension acts as a flexible tether essential for the assembly of IFT trains at the ciliary base.

While the meticulous design and commendable results make this paper a significant contribution, especially in elucidating the role of unstructured tethers in IFT train interactions, several points require further clarification:

1. Assembly of Dynein-2 in Double Knockouts: The paper could be enhanced by addressing how dynein-2 assembles when both WDR34 and WDR60 are knocked out. Failure to do so could call into question some of the conclusions about double knockout effects and the role of the N-terminal tether. Negative staining analysis might be one way to investigate this.

Thank you for this suggestion. As mentioned in response to Reviewer #1, point 2, we don't wish to suggest that loss of WDR34 and WDR60 has no effect on dynein-2 assembly. Indeed, we have shown using negative stain EM that loss of WDR34 and WDR60 weakens DYNC2H1 self-assembly (Figure 3b of Toropova *et al.*, 2019). Importantly, however, our experiments reveal that appending the N-terminal extension of WDR60 onto WDR34 enables remarkably full levels of dynein-2 function in the absence of WDR60. Similarly, deletion of WDR34 has only a mild effect on dynein-2 cellular function. Thus, dynein-2 is strikingly resilient to the loss of either the WDR34 or WDR60 β -propeller domain and light-chain binding sites, provided at least one IC with the WDR60 N-terminal extension is present. These results indicate that contributions of the WDR34 or WDR60 β -propeller domain and light-chain binding sites to dynein-2 assembly are secondary to the role of the flexible WDR60 N-terminal region in targeting dynein-2 to cilia. We have added this clarification to the Discussion.

2. Role of WDR60's N-terminal Extension in Ciliogenesis: The paper raises the question of why ciliogenesis is not abolished when WDR60 is knocked out, despite its significant role in the dynein-2 mechanism. Addressing this inconsistency would strengthen the paper's conclusions. Also, it

might be very intriguing if this tether can be transplanted to other non-cilia proteins for ciliary targeting. It will be great if the authors can test it.

We apologize our interpretation was not sufficiently clear here. Based on our data and model, we would not expect ciliogenesis to be abolished when WDR60 is knocked out. When WDR60 is absent, we observe a much reduced but still detectable level of dynein-2 and retrograde IFT in the cilium, sufficient to support ciliogenesis. This is consistent with the structural model of dynein-2 binding to the anterograde IFT train, which involves interactions made by both DYNC2H1 and the WDR60 N-terminal extension. When WDR60 is knocked out, the latter is abolished but the former can persist. In contrast, when both WDR34 and WDR60 are absent, we do not detect dynein-2 in the cilium and ciliogenesis is severely perturbed. It is likely that WDR34 stabilizes DYNC2H1 in the bent conformation required for IFT train binding. Thus, WDR34 can support a minimal level of dynein-2 function in the absence of WDR60 and its N-terminal region, consistent with our structural model. We have added a new figure panel to clarify our interpretation (Figure 6C), which we think has improved the manuscript. Regarding using the IFT-B-binding regions of WDR60 to target non-ciliary protein to cilia, this is a very interesting idea. As the binding sites are likely to be individually weak and synergize through multi-valency, we think they may not be the most effective ciliary targeting sequences individually, but there is nice evidence in *C. elegans* that the entire N-terminal sequence of the WDR60 homolog is sufficient to target GFP to phasmid cilia (Fig. 2B,C of De-Castro *et al.*, 2022), in agreement with the Reviewer's suggestion.

3. Validation of AlphaFold Predictions: Although AlphaFold is a powerful tool for protein structure prediction, it cannot replace the need for empirical evidence. Biochemical assays validating the key residues as indicated by AlphaFold would lend greater credence to the findings.

As described in our response to Reviewer #1, point 1, we have performed a pulldown assay with purified proteins to confirm the direct interaction between the N-terminal region of WDR60 (residues 355-449) and the CH domain of IFT54 in IFT-B (new Appendix Figure S4A). These data provide biochemical evidence for direct WDR60:IFT-B interaction.

4. Statistics: The paper would benefit from clearly indicating the p-values on figures to allow for a robust statistical interpretation.

If acceptable, we would prefer to keep the P-values for the numerous comparisons in the figures legends as in the original submission, to avoid cluttering the figures while still enabling robust statistical interpretation.

Referee #3:

*Structure and tethering mechanism of Dynein-2's heterodimeric intermediate chains
Mukhopadhyay et al.*

Dyneins are the main motors responsible for anterograde transport along microtubules and are far more complex than the other cytoskeletal motors, kinesins and myosins. In cilia and flagella, this anterograde movement is powered by dynein-2. Although dynein-2 is a homodimer of identical heavy chains, it is unusual in that it contains two different intermediate chains-WDR34 and WDR60-in contrast to its cytoplasmic counterpart, which contains two copies of the same polypeptide. Symmetry mismatches are always interesting in structural biology, and this one is even more so given the regulatory role played by the dynein-2 intermediate chains.

In this manuscript, Mukhopadhyay and colleagues use a combination of cryo-EM, crystallography, and cell-based functional assays to reveal how WDR34 and WDR60 interact with each copy of the dynein heavy chain in different ways and how the two intermediate chains have different functional

roles. The most interesting result (in this reviewer's opinion) was that the long N-terminal extension of WDR60 is involved in the targeting of dynein-2 to cilia, and that its function could be preserved by transferring it to WDR34 in a cell lacking WDR60.

This is a very well written manuscript presenting an important contribution to our understanding of a complex motor with an even more complex regulation. The data will be of interest to anyone studying dynein, as well as to those studying cilia biology. I have no major concerns with the manuscript. I outline a couple of minor suggestions below that I believe could further strengthen it.

- Figure 1H-J. The authors state that while previous data suggested that DYNLT1 and DYNLT2B could form a heterodimer, a dimer of dimers could not be ruled out. The crystal structure presented in Fig. II clearly shows that DYNLT1 and DYNLT2B can form a heterodimer. Would it be possible to dock this structure into the density shown in Fig. 1H to rule out the possibility that a dimer of heterodimers can account for it? I can't tell whether the resolution in that region of the map isn't good enough to do this in an unambiguous manner, but if it is, it could put this (minor) issue to rest.

Thank you for this suggestion. As shown in a new figure panel (Appendix Figure S2E), the DYNLT1-DYNLT2B crystal structure accounts for the unfilled density in the cryo-EM map. We have highlighted this helpful point in the text.

- While I agree that the most parsimonious conclusion for the data presented is that "the N-terminal extension of WDR60 is a modular element that can be transplanted onto WDR34 to rescue function", I am not 100% convinced that the data available out there rule out the possibility that one of the two WDR variants could occupy both binding sites if the other one is absent from the cell. If that were the case, then it would be the WD40s that are "modular", while the N-terminal extension of WDR60 would be found in its normal place.

Since I find this possibility far less likely than the one proposed by the authors, I would not expect them to do any additional experiments to address it. But I was wondering if something simple, such as a few runs with AlphaFold Multimer, could provide additional support for their proposal. For example, can AlphaFold Multimer come up with WDR34-WDR34 or WDR60-WDR60 homodimers that could still fit into the dynein-2 complex? I find the statement in the Discussion that "WDR34 and WDR60 cannot homodimerize" a bit too strong given the papers cited. I agree that those papers showed that WDR34 and WDR60 did not dimerize under their conditions and with the approaches used, but "did not homodimerize" is quite different from "cannot homodimerize". Similarly, would it be possible to run AlphaFold Multimer using either two copies of WDR34 or of WDR60 plus the surrounding region of dynein's heavy chain? Even if absence of evidence (i.e. if AlphaFold cannot come up with structures in any of those cases) is not evidence of absence, this would help me further dismiss my alternative hypothesis. If none of these suggestions work, I would be satisfied if the Discussion acknowledged that the possibility that one WDR can occupy both binding sites, however unlikely, has not been completely ruled out.

We agree our wording on WDR34 or WDR60 homodimerization was heavy handed. We have rephrased to: "*WDR34 and WDR60 have not been found to homodimerize.*" We have also performed the suggested AlphaFold predictions of WDR34-WDR34 and WDR60-WDR60 homodimers interacting with DYNC2H1. The predictions do not match the IFT-train-binding conformation of the dynein-2 complex, as they lack the asymmetry that is a key feature of dynein-2 when bound to IFT-B. That said, we agree that it is difficult to completely rule out that two copies of WDR34 (or WDR60) could occupy both DYNC2H1 sites transiently or in an infrequent fashion. To us, this possibility is not at odds with the statement: "*the N-terminal extension of WDR60 is a modular element that can be transplanted onto WDR34 to rescue function.*" Here, we mean that the WDR60 N-terminal extension is modular in the sense that

it can function when appended to WDR34 and does not depend on the WDR60 β -propeller domain or light-chain binding sites. We hope this clarifies.

Referee #4:

Mukhopadhyay et al. reported the structure and regulatory mechanisms of IFT-dynein by its two distinct intermediate chains WDR34 and WDR60. IFT-dynein powers retrograde IFT and its dysfunction causes many types of ciliary diseases; however, its regulatory mechanism remains unclear. Using cryo-EM and other structure biological approaches, this study shows that the WDR60 β -propeller uses a unique insert in its third blade to stabilize DYNC2H1 in a straight confirmation and that WDR34 uses a classic β -propeller mode interaction to engage DYNC2H1. They showed that the WDR34 and WDR60 β -propellers are held together by light chains. Their data suggest that the N-proximal region of WDR60 may act as a flexible tether to IFT-particles, providing additional insights for the ciliary entry and anterograde transport of IFT-dynein. Using Cas9-based knockout and imaging tools, this study revealed that WDR34 and WDR60 have redundant function in the formation of cilia. They showed that WDR60 appears to play a more important role likely because its N-terminal extension facilitates ciliary targeting of dynein-2. Many of the experiments are well performed. In particular, the visualization of dynein-2 heavy chain is not trivial but very informative. The manuscript is nicely written. While the characterization of detailed interaction of dynein-2's heavy chain, associated chains, and IFT-particles appears to be incremental, the idea of such a unique tethering mechanism is important and intriguing. I support its publication in EMBO J as long as the following minor issues can be addressed through discussion and rewriting.

1. The author can add one paragraph to discuss whether and how missense mutations of WDR34 and WDR60 are associated with any types of ciliopathies. Are there any mutations detected around the tethering interface?

We recently published an analysis of ciliopathy-associated missense mutations in WDR34 (Shak *et al.*, 2023). To date no ciliopathy-associated missense mutations have been reported in the tethering interfaces of WDR60. Interestingly, mutation of DYNC2H1 R587 (which our cryo-EM structure shows to interact with both WDR34 and WDR60) is associated with severe short-rib thoracic dysplasia (Merrill *et al.*, 2009). We have added this point to the manuscript.

2. How common is the redundance of WDR34 and WDR60 in other ciliated organisms? Green algae or nematodes

The two intermediate chains of dynein-2 were discovered in *C. reinhardtii* as FAP133 and FAP163 (WDR34 and WDR60 orthologs respectively). The extent of redundancy between FAP133 and FAP163 in *C. reinhardtii* is not yet known. Intriguingly, in *C. elegans*, it is unclear if a WDR34 homolog exists or if the *C. elegans* WDR60 homolog functions as a homodimer (Higashida & Niwa, 2022). We highlight this point in the Introduction.

3. The model figure 6 can be expended to illustrate the consequence of WDR34 or WDR60 KOs.

Thank you for this suggestion. We have added the impact of the KOs to Figure 6, which we think has improved the manuscript.

Dear Anthony,

Thank you for submitting a revised version of your manuscript. Your study has now been seen by three of the original referees, who find that their previous concerns have been addressed and now recommend acceptance of the manuscript.

There now remain only a few editorial points that need addressing before I can extend acceptance of the manuscript:

1. Please refer to the figure panels 6A-C in the manuscript text.
2. In the Appendix, please add page numbers to the table of contents
3. Please update the Appendix figure nomenclature to Appendix Figure S1-S5 and Appendix Table S1-S3 in the table of contents and figure/table legends.
4. In the Data Availability section, please add resolvable links to the datasets. More information about the format of this section can be found here: <https://www.embopress.org/page/journal/14602075/authorguide#dataavailability>.
5. Our data editors have flagged the following issues in figure legends that need correcting:
 - Please indicate the statistical test used for data analysis in the legend of figure 5i.
 - Please note that the error bars are not defined in the legends of figures 5g, i.
6. Papers published in The EMBO Journal are accompanied online by a 'Synopsis' to enhance discoverability of the manuscript. It consists of A) a short (1-2 sentences) summary of the findings and their significance, B) 3-4 bullet points highlighting key results and C) a synopsis image that is 550x300-600 pixels large (width x height, jpeg or png format). You can either show a model or key data in the synopsis image. Please note that the image size is rather small and that text needs to be readable at the final size. Please send us this information together with the revised manuscript.

With best wishes,

Ieva

We realize that it is difficult to revise to a specific deadline. In the interest of protecting the conceptual advance provided by the work, we recommend a revision within 3 months (14th Apr 2024). Please discuss the revision progress ahead of this time with the editor if you require more time to complete the revisions.

Referee #1:

The authors have now addressed all of my concerns. Most importantly, they now present new data verifying a direct interaction between WDR60 and the CH domain of IFT54.

I recommend publication.

Referee #2:

The authors clearly addressed all my questions, and I strongly recommend publication of this paper in EMBO J.

Referee #3:

I am satisfied with how the authors addressed my prior concerns and support publication of the manuscript.

The authors addressed the remaining editorial issues.

Dear Anthony,

Thank you for addressing the final editorial issues. I am now pleased to inform you that your manuscript has been accepted for publication.

I will look into the synopsis text in the next couple of days and let you know if any edits to the journal style are needed.

If you have any questions, please do not hesitate to contact the Editorial Office. Thank you for this contribution to The EMBO Journal and congratulations on a great paper!

Best wishes,

leva

leva Gailite, PhD
Senior Scientific Editor
The EMBO Journal
Meyerhofstrasse 1
D-69117 Heidelberg
Tel: +4962218891309
i.gailite@embojournal.org
